# Prompt Tuning for CLIP on the Pretrained Manifold

Xi Yang [1 2]   Yuanrong Xu [* 2]   Weigang Zhang [2]   Guangming Lu [3]   David Zhang [4]   Jie Wen [* 1 3]

## Abstract

Prompt tuning introduces learnable prompt vectors that adapt pretrained vision-language models to downstream tasks in a parameter-efficient manner. However, under limited supervision, prompt tuning alters pretrained representations and drives downstream features away from the pretrained manifold toward directions that are unfavorable for transfer. This drift degrades generalization. To address this limitation, we propose ManiPT, a framework that performs prompt tuning on the pretrained manifold. ManiPT introduces cosine consistency constraints in both the text and image modalities to confine the learned representations within the pretrained geometric neighborhood. Furthermore, we introduce a structural bias that enforces incremental corrections, guiding the adaptation along transferable directions to mitigate reliance on shortcut learning. From a theoretical perspective, ManiPT alleviates overfitting tendencies under limited data. Our experiments cover four downstream settings: unseen-class generalization, few-shot classification, cross-dataset transfer, and domain generalization. Across these settings, ManiPT achieves higher average performance than baseline methods. Notably, ManiPT provides an explicit perspective on how prompt tuning overfits under limited supervision.

## 1. Introduction

Recent large-scale pretrained vision-language models (VLMs) such as CLIP (Radford et al., 2021), ALIGN (Jia et al., 2021), and Florence (Yuan et al., 2021) leverage massive image and text-aligned data to learn general representations. These models support open-vocabulary recognition

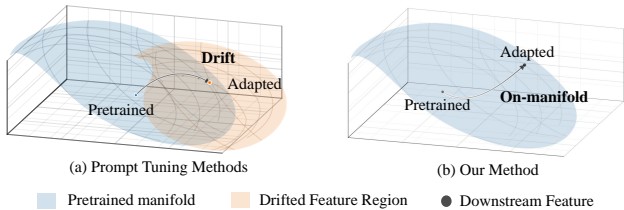

*Figure 1.* Manifold drift and manifold preservation on the CLIP feature space. (a) **Prompt tuning.** Under limited supervision, prompt tuning drives adapted representations away from the pretrained manifold. (b) **ManiPT.** The proposed method constrains adapted representations to stay close to the pretrained manifold.

and transfer (Radford et al., 2021; Gu et al., 2021) and are widely adopted as foundation models for downstream tasks. However, under practical constraints such as label scarcity or domain shifts, direct full fine-tuning incurs high computational costs and risks compromising the structure of pretrained representations. By freezing the pretrained backbone and introducing a small number of learnable vectors to modulate model behavior, prompt tuning maintains strong performance on downstream benchmarks while significantly reducing the number of trainable parameters and is now a common paradigm for adapting VLMs.

CLIP establishes a transferable feature manifold via large-scale pretraining. Under limited supervision, however, prompt learning tends to exploit local discriminative signals. These signals contain some transferable semantics but also capture spurious correlations that are valid only within the limited training data, such as background patterns or texture artifacts. Although this strategy ensures separability on in-domain data, it causes adapted representations to deviate from the pretrained manifold, as illustrated in Figure 1. Without the geometric support of the large-scale pretrained manifold, the adapted representations fail to generalize reliably to unseen classes or cross-domain distributions, which impairs overall generalization performance.

While existing prompt-based adaptation methods mitigate overfitting by making prompts more expressive or adding heuristic regularization, they seldom explicitly control how prompt updates change representations relative to the frozen CLIP features. For example, CoOp and CoCoOp (Zhou et al., 2022b;a) improve transfer by learning class- or instance-conditioned prompts. MaPLe (Khattak et al., 2023a) and related deep prompting methods (Jia et al., 2022)

[1]Guizhou University, Guiyang, China [2]Harbin Institute of Technology, Weihai, China [3]Harbin Institute of Technology, Shenzhen, China [4]The Chinese University of Hong Kong, Shenzhen, China. Correspondence to: Yuanrong Xu <xuyuanrong@hit.edu.cn>, Jie Wen <jiewen_pr@126.com>.

*Proceedings of the 43rd International Conference on Machine Learning*, Seoul, South Korea. PMLR 306, 2026. Copyright 2026 by the author(s).

inject prompts into multiple layers to increase capacity, and CLIP-Adapter (Gao et al., 2024) attaches shallow adapters onto frozen CLIP to bridge domain gaps. Regularization-oriented approaches such as PromptSRC (Khattak et al., 2023b) and CoPrompt (Roy & Etemad, 2023) introduce additional losses that aim to stabilize training, but these penalties are typically defined on logits, parameters, or prompt content rather than directly constraining feature geometry.

In ManiPT, we instead treat the frozen CLIP representations as proxies for the pretrained manifold. By enforcing feature-level cosine consistency with these frozen references on both the visual and textual sides, ManiPT confines the learned representations within the pretrained geometric neighborhood. However, simply staying within the neighborhood does not guarantee transferability, as shortcut solutions may still exist locally. To address this, ManiPT introduces a structural bias through normalized additive aggregation. This design enforces incremental corrections, guiding the representations to adjust along transferable directions while suppressing reliance on dataset-specific shortcuts.

Building on this analysis, we propose ManiPT, a framework for prompt tuning on the pretrained manifold, shown in Figure 1. Our contributions are summarized below:

- We identify manifold drift as a critical factor limiting generalization under limited supervision and propose ManiPT to mitigate the resulting degradation.

- We introduce cosine consistency constraints to prevent manifold drift and a structural bias to mitigate shortcut learning through incremental corrections.

- We provide theoretical guarantees on generalization error and demonstrate that ManiPT consistently outperforms baseline methods across extensive experiments.

## 2. Related Work

**Manifold Learning.** Manifold learning (Tenenbaum et al., 2000) is built on the assumption that high-dimensional observations often lie approximately on a low-dimensional nonlinear manifold. Early studies (Roweis & Saul, 2000) mainly focus on recovering global geometric structure from local neighborhood relations and use this structure for nonlinear dimensionality reduction and representation learning. Building on this line of work, manifold regularization (Belkin et al., 2006) explicitly injects a geometric prior of smoothness along the data manifold into the learning objective and, in settings such as semi-supervised learning, uses graph-based regularization to constrain variation at unlabeled samples, which leverages the geometry of the marginal distribution to improve generalization. With the rise of deep learning, manifold ideas expand from explicit embeddings to geometric constraints in latent spaces. Contractive Auto-Encoders (Rifai et al., 2011) encourage representations to

be sensitive to variations along tangent directions of the manifold while remaining robust to perturbations in directions orthogonal to the manifold, so that local manifold structure can be captured. Manifold Mixup (Verma et al., 2019) performs interpolation in hidden layers to smooth the decision boundary and to learn intermediate representations that transfer well across tasks. In the same spirit, we treat the pretrained CLIP feature space as a robust manifold. Our goal extends beyond merely preserving this structure. We aim to constrain the feature adaptation within the manifold neighborhood while biasing the adaptation toward transferable directions along the manifold surface.

**Prompt Tuning.** Early attempts such as CoOp (Zhou et al., 2022b) substitute handcrafted templates with learnable context vectors. To address limited generalization on unseen classes, subsequent approaches such as Co-CoOp (Zhou et al., 2022a) condition prompts on image instances. MaPLe (Khattak et al., 2023a) enhances modality alignment through hierarchical prompting. DPC (Li et al., 2025a) decouples prompts into dual branches for specialization. DePT (Zhang et al., 2024) isolates channel-wise biases to reduce interference. To mitigate overfitting and catastrophic forgetting, recent studies introduce regularization strategies that maintain the discriminative power of the backbone, including PromptSRC (Khattak et al., 2023b) and CoPrompt (Roy & Etemad, 2023), which we use as strong baselines. More recent research focuses on integrating prior knowledge and external information. TAC (Hao et al., 2025) exploits task-aware priors to generate task contexts, and Text-Refiner (Xie et al., 2025) utilizes internal visual knowledge to refine prompts. Approaches such as LLaMP (Zheng et al., 2024), TAP (Ding et al., 2024), and ATPrompt (Li et al., 2025b) leverage external large language models (LLMs) to enrich semantic descriptions and structured attributes of prompts. Existing methods primarily enhance adaptation capacity by increasing prompt plasticity, which often necessitates heuristic regularization to mitigate the resulting overfitting. Distinct from these approaches, ManiPT adopts a geometric perspective. We introduce consistency constraints to confine the learned representations within the pretrained geometric neighborhood and a structural bias to enforce incremental corrections. This dual mechanism ensures that the model not only stays on the manifold but also guides the feature adaptation along transferable directions while suppressing dataset-specific shortcuts.

## 3. Preliminaries and Insights

We follow the standard CLIP-based protocol (Radford et al., 2021; Zhou et al., 2022b) for image classification. The main symbols and notation used throughout this section and the subsequent analysis are summarized in Appendix F.

**CLIP-based Image Classification.** Given a label set $\mathcal{C} = \{1, \ldots, C\}$ and an image $\mathbf{x}$, CLIP maps texts and

images into a shared space via frozen encoders $\mathcal{T}$ and $\mathcal{V}$. For class $c$ with template $\mathbf{t}_c$, the normalized text feature is $\mathbf{z}_c^{\text{txt}} = \mathcal{T}(\mathbf{t}_c)/\|\mathcal{T}(\mathbf{t}_c)\| \in \mathbb{R}^d$, where $d$ is the embedding dimension. Similarly, the normalized visual feature is $\mathbf{z}_{\mathbf{x}}^{\text{vis}} = \mathcal{V}(\mathbf{x})/\|\mathcal{V}(\mathbf{x})\| \in \mathbb{R}^d$. We compute logits $\ell_c(\mathbf{x}) = \tau(\mathbf{z}_{\mathbf{x}}^{\text{vis}})^{\top}\mathbf{z}_c^{\text{txt}}$ with temperature $\tau$, and the predicted label is $\hat{y}(\mathbf{x}) = \arg\max_c \ell_c(\mathbf{x})$. The probabilistic prediction is:

$$p(c \,|\, \mathbf{x}) = \frac{\exp(\ell_c(\mathbf{x}))}{\sum_{j=1}^{C} \exp(\ell_j(\mathbf{x}))}. \quad (1)$$

**Prompt Tuning on CLIP.** Prompt tuning adapts CLIP to a downstream task by freezing the pretrained text encoder $\mathcal{T}$ and image encoder $\mathcal{V}$ and learning a small set of prompt parameters that modulate the representations.

For each class $c$, prompt tuning replaces the handcrafted context in the class template with $m$ learnable context vectors. Let $\mathbf{P}_0 = \{\mathbf{p}_1, \ldots, \mathbf{p}_m\}$ denote the learnable context. The prompted text sequence is denoted by $\mathbf{t}_c(\mathbf{P}_0)$, and the resulting class text feature is:

$$\mathbf{h}_c^{\text{txt}} = \frac{\mathcal{T}(\mathbf{t}_c(\mathbf{P}_0))}{\|\mathcal{T}(\mathbf{t}_c(\mathbf{P}_0))\|}. \quad (2)$$

Beyond input-level context, we inject learnable prompts into intermediate Transformer (Vaswani et al., 2017) layers. We denote the deep prompts for the text encoder as $\{\mathbf{P}_k\}_{k=1}^{K-1}$, where $K$ is the number of layers. Collecting all text-side prompts as $\mathcal{P}^{\text{txt}} = \{\mathbf{P}_0, \mathbf{P}_1, \ldots, \mathbf{P}_{K-1}\}$, the prompted text feature can be written as:

$$\mathbf{h}_c^{\text{txt}} = \frac{\mathcal{T}(\mathbf{t}_c, \mathcal{P}^{\text{txt}})}{\|\mathcal{T}(\mathbf{t}_c, \mathcal{P}^{\text{txt}})\|}. \quad (3)$$

On the visual side, prompt vectors can also be introduced into the image encoder. We denote the input-level visual prompts by $\tilde{\mathbf{P}}_0$ (inserted after patch embedding) and the deep visual prompts by $\{\tilde{\mathbf{P}}_k\}_{k=1}^{K-1}$. Let $\mathcal{P}^{\text{vis}} = \{\tilde{\mathbf{P}}_0, \tilde{\mathbf{P}}_1, \ldots, \tilde{\mathbf{P}}_{K-1}\}$ be all visual prompts. The prompted image feature is then:

$$\mathbf{h}_{\mathbf{x}}^{\text{vis}} = \frac{\mathcal{V}(\mathbf{x}, \mathcal{P}^{\text{vis}})}{\|\mathcal{V}(\mathbf{x}, \mathcal{P}^{\text{vis}})\|}. \quad (4)$$

With prompt-adapted features, CLIP performs classification by cosine similarity matching between $\mathbf{h}_{\mathbf{x}}^{\text{vis}}$ and the class text features $\mathbf{h}_c^{\text{txt}}$. We compute:

$$\ell_c(\mathbf{x}) = \tau \cdot \left(\mathbf{h}_{\mathbf{x}}^{\text{vis}}\right)^{\top} \mathbf{h}_c^{\text{txt}}, \quad (5)$$

followed by a softmax over classes to obtain $p(c \,|\, \mathbf{x})$.

**Representing Manifold Drift and Shortcut Reliance.** Based on the manifold hypothesis (Tenenbaum et al., 2000), we posit that CLIP pretrained features exhibit a low-dimensional geometric structure despite residing in a high-dimensional space. As this intrinsic structure is not explicitly accessible, we utilize the principal component analysis (PCA) subspace (Abdi & Williams, 2010) of the pretrained feature point cloud as a computable approximation to quantify the manifold shift induced by prompt tuning. Specifically, let $\mathbf{Z} \in \mathbb{R}^{N \times d}$ and $\mathbf{H} \in \mathbb{R}^{N \times d}$ denote the normalized feature point clouds of the pretrained and prompt-tuned models respectively, where $N$ is the number of samples and $d$ is the feature dimension. Let $\boldsymbol{\mu}$ be the mean of the pretrained feature cloud, and let $\bar{\mathbf{Z}}$ and $\bar{\mathbf{H}}$ denote the pretrained and prompt-tuned features centered by this common mean. We perform PCA on $\bar{\mathbf{Z}}$ and let $\mathbf{V}^{\text{pca}} \in \mathbb{R}^{d \times d^{\text{pca}}}$ collect the top-$d^{\text{pca}}$ principal directions. We define the projection matrix $\mathbf{P}^{\text{pca}} = \mathbf{V}^{\text{pca}}(\mathbf{V}^{\text{pca}})^{\top} \in \mathbb{R}^{d \times d}$ onto this principal subspace, and let $\mathbf{I}_d \in \mathbb{R}^{d \times d}$ be the identity matrix. Then the approximate manifold shift is formulated as follows. See Appendix D.1 for detailed definitions.

$$\Delta = \frac{\|\bar{\mathbf{H}}(\mathbf{I}_d - \mathbf{P}^{\text{pca}})\|_F^2}{\|\bar{\mathbf{H}}\|_F^2} - \frac{\|\bar{\mathbf{Z}}(\mathbf{I}_d - \mathbf{P}^{\text{pca}})\|_F^2}{\|\bar{\mathbf{Z}}\|_F^2}, \quad (6)$$

where $\|\cdot\|_F$ denotes the Frobenius norm. $\Delta > 0$ signifies prompt-tuned features drift away from the principal subspace of the pretrained CLIP model, increasing the risk of overfitting to dataset-specific biases.

Consider the decomposition of the feature space into a transferable subspace $\mathcal{Q}$ supported by pretraining and a shortcut subspace $\mathcal{S}$ containing dataset-specific variations. Let $\boldsymbol{\delta}$ denote the feature shift induced by prompt tuning. We formulate the decomposition as:

$$\boldsymbol{\delta} = \boldsymbol{\delta}^{\mathcal{Q}} + \boldsymbol{\delta}^{\mathcal{S}}, \quad (7)$$

where $\boldsymbol{\delta}^{\mathcal{Q}}$ resides in $\mathcal{Q}$ and represents refinement along robust semantic directions while $\boldsymbol{\delta}^{\mathcal{S}}$ belongs to $\mathcal{S}$ and denotes deviation into the shortcut subspace. When the training set is small, minimizing empirical risk tends to exploit discriminative signals present in $\mathcal{S}$ such as background noise or texture biases. These signals are sufficient to separate the limited support set but lack semantic universality. Under limited supervision, optimization tends to amplify $\boldsymbol{\delta}^{\mathcal{S}}$, causing the learned representations to diverge from the robust semantic directions supported by the pretrained manifold.

**A Framework for Performing Prompt Tuning on the Pretrained Manifold.** Building on the analysis in Eq. (6) and Eq. (7), we propose a framework to perform prompt tuning on the pretrained manifold. ManiPT imposes cosine consistency constraints to confine the learned representations within the pretrained geometric neighborhood. However, since shortcut components $\boldsymbol{\delta}^{\mathcal{S}}$ may still persist within the manifold neighborhood, we further introduce a structural bias to enforce incremental corrections, thereby biasing the adaptation toward transferable directions.

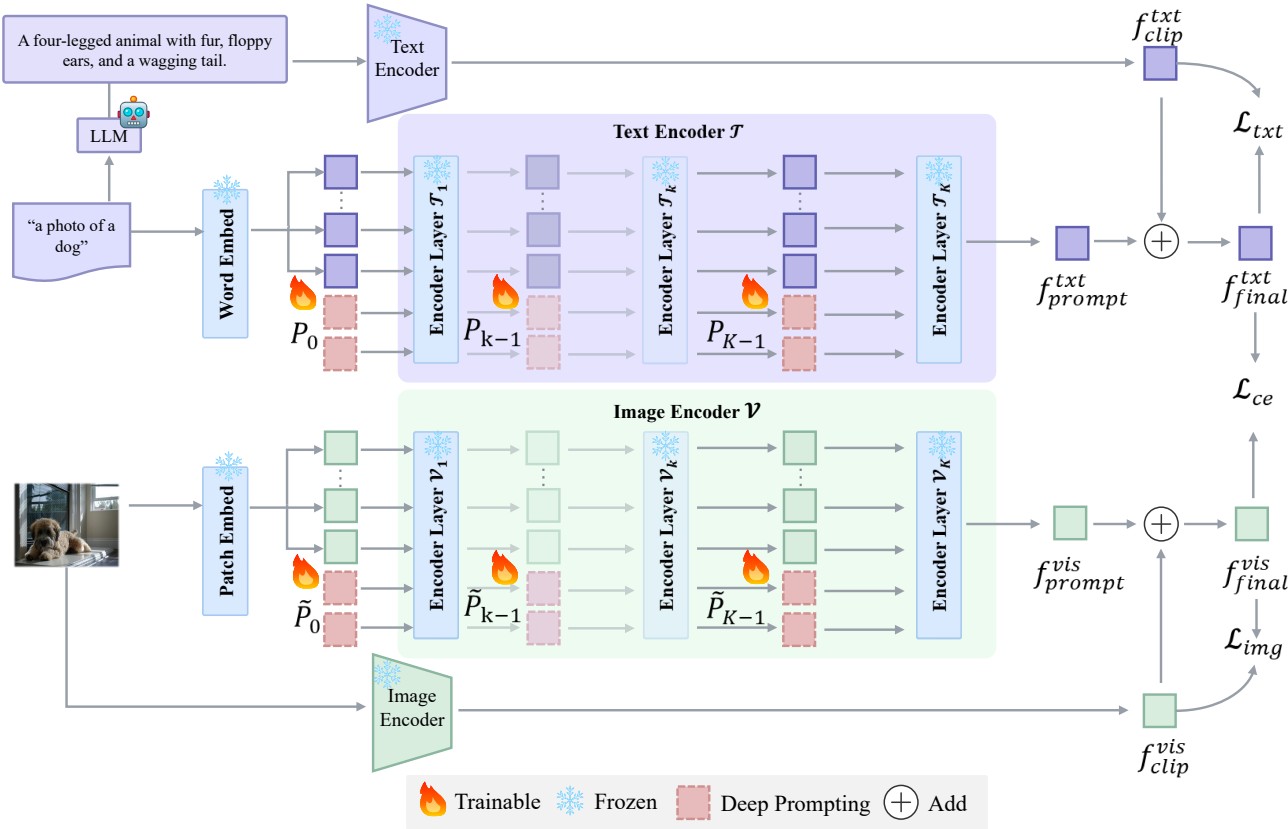

*Figure 2.* Overview of ManiPT. The framework enriches class descriptions with an LLM, constructs a text feature bank as semantic prototypes, and applies cosine consistency constraints together with a structural bias to keep prompt tuning near the pretrained manifold.

# 4. Prompt Tuning on the Pretrained Manifold

This section presents ManiPT and components from Section 3. Figure 2 shows the pipeline. We describe LLM-based knowledge enrichment, cosine consistency constraints, and the structural bias. Finally, we show that when empirical risks are comparable, ManiPT can theoretically attain a lower population risk than standard prompt tuning.

## 4.1. LLM-based Knowledge Enrichment

To reduce shortcut semantics learned under few-shot supervision, we construct stable semantic reference texts for each class and encode them into a text feature bank serving as a semantic prototype for subsequent constraints on the pretrained manifold. For each class $c \in \mathcal{C}$, we query an LLM to generate descriptions, yielding a description set $A_c = \{a_{c,j}\}_{j=1}^n$, where $n$ is the count per class, and aggregate these as $A = \bigcup_{c \in \mathcal{C}} A_c$. For efficiency, we pre-encode these descriptions into normalized CLIP text features:

$$E_c = \left\{ \frac{\mathcal{T}(a)}{\|\mathcal{T}(a)\|} \;\middle|\; a \in A_c \right\}. \tag{8}$$

## 4.2. Cosine Consistency Constraints

As shown in Eq. (6) and Eq. (7), the feature change induced by prompt tuning decomposes into a component $\boldsymbol{\delta}^{\mathcal{Q}}$ in the

transferable subspace and a shortcut component $\boldsymbol{\delta}^{\mathcal{S}}$ in the non-transferable subspace. Under limited supervision, optimization is more prone to amplifying $\boldsymbol{\delta}^{\mathcal{S}}$, causing manifold drift ($\Delta > 0$). Since CLIP makes predictions based on cosine similarity between normalized features, we impose cosine consistency constraints on both modalities. This explicitly confines the feature adaptation within the pretrained geometric neighborhood, preventing significant deviation from the valid support of the pretrained manifold.

**Visual-side Consistency Constraint.** For sample $\mathbf{x}_i$, let $\mathbf{z}_{\mathbf{x}_i}^{\mathrm{vis}}$ and $\mathbf{h}_{\mathbf{x}_i}^{\mathrm{vis}}$ be the normalized visual features from the frozen and prompt-tuned branches. We define the visual-side consistency loss as:

$$\mathcal{L}^{\mathrm{img}} = \frac{1}{N} \sum_{i=1}^N \left( 1 - \left\langle \mathbf{h}_{\mathbf{x}_i}^{\mathrm{vis}}, \mathbf{z}_{\mathbf{x}_i}^{\mathrm{vis}} \right\rangle \right). \tag{9}$$

This constraint explicitly limits the geometric deviation of the adapted features from the frozen representations.

**Text-side Consistency Constraint.** To obtain a more stable semantic reference, we leverage the LLM description feature set $E_c$ constructed in Section 4.1. For each class $c$, we aggregate it into a normalized semantic prototype $\mathbf{w}_c = (\sum_{\boldsymbol{\eta} \in E_c} \boldsymbol{\eta})/\|\sum_{\boldsymbol{\eta} \in E_c} \boldsymbol{\eta}\|$. Let $\mathbf{h}_c^{\mathrm{txt}}$ be the normal-

ized prompt-tuned text feature. The text-side consistency loss is:

$$\mathcal{L}^{\text{txt}} = \frac{1}{C} \sum_{c \in \mathcal{C}} \left( 1 - \langle \mathbf{h}_c^{\text{txt}}, \mathbf{w}_c \rangle \right). \qquad (10)$$

Unlike the visual constraint, we incorporate LLM-derived semantic priors. This ensures the prompted text features remain semantically anchored to robust class descriptions, mitigating the risk of semantic drift.

### 4.3. Structural Bias

While consistency constraints confine the learned representations within a geometric neighborhood, this is a necessary but not sufficient condition for transferability, as shortcut solutions may still exist locally. To address this, we further introduce a structural bias. Specifically, we explicitly preserve the frozen CLIP representations in the final features used for classification. This design restricts prompt learning to incremental corrections on top of the pretrained manifold.

For sample $\mathbf{x}$ and class $c$, we obtain normalized frozen features $\mathbf{z}_{\mathbf{x}}^{\text{vis}}, \mathbf{z}_c^{\text{txt}}$ and prompt features $\mathbf{h}_{\mathbf{x}}^{\text{vis}}, \mathbf{h}_c^{\text{txt}}$. We aggregate them residually and renormalize to produce final classification features:

$$\mathbf{f}_{\mathbf{x}}^{\text{vis}} = \frac{\mathbf{z}_{\mathbf{x}}^{\text{vis}} + \mathbf{h}_{\mathbf{x}}^{\text{vis}}}{\|\mathbf{z}_{\mathbf{x}}^{\text{vis}} + \mathbf{h}_{\mathbf{x}}^{\text{vis}}\|},$$
$$\mathbf{f}_c^{\text{txt}} = \frac{\mathbf{z}_c^{\text{txt}} + \mathbf{h}_c^{\text{txt}}}{\|\mathbf{z}_c^{\text{txt}} + \mathbf{h}_c^{\text{txt}}\|}. \qquad (11)$$

This additive structure induces a geometric contraction, which imposes a hard constraint on the optimization behavior. Instead of allowing arbitrary deviations, this design enforces incremental corrections. By anchoring the final representations to the frozen representations, the structural bias effectively biases the adaptation toward transferable directions implied by the pretraining, thereby reducing reliance on dataset-specific shortcut components.

**Classification Objective.** Based on the fused final representations, we follow CLIP's cosine similarity matching form to compute the logit for each class $c \in \mathcal{C}$:

$$\ell_c(\mathbf{x}) = \tau \cdot \left( \mathbf{f}_{\mathbf{x}}^{\text{vis}} \right)^\top \mathbf{f}_c^{\text{txt}}. \qquad (12)$$

We optimize prompt parameters using cross-entropy loss:

$$\mathcal{L}^{\text{ce}} = -\frac{1}{N} \sum_{i=1}^{N} \log \frac{\exp(\ell_{y_i}(\mathbf{x}_i))}{\sum_{c \in \mathcal{C}} \exp(\ell_c(\mathbf{x}_i))}. \qquad (13)$$

**Training Objective.** For conceptual clarity, we formulate the consistency constraints in this section using the intermediate prompt-adapted features. However, to leverage the structural bias defined in Eq. (11), our final training objective applies these constraints specifically to the fused

representations $\mathbf{f}_{\mathbf{x}}^{\text{vis}}$ and $\mathbf{f}_c^{\text{txt}}$. This design ensures that the actual representations used for decision-making remain geometrically aligned with the pretrained representations.

Consequently, the total objective function is formulated to minimize the classification error while satisfying these manifold consistency constraints:

$$\mathcal{L}^{\text{total}} = \mathcal{L}^{\text{ce}} + \lambda \left( \mathcal{L}^{\text{img}} + \mathcal{L}^{\text{txt}} \right), \qquad (14)$$

where $\lambda$ balances regularization strength. Optimizing this objective on structurally biased features guides the model to discover transferable directions within the pretrained manifold. The pseudocode of ManiPT is summarized in Appendix A.3, and training and inference costs are compared in Appendix C.6. ManiPT incurs slightly higher latency than single-branch methods due to dual-branch fusion but remains suitable for real-time deployment.

### 4.4. Theoretical Analysis

We develop the core theoretical guarantees for ManiPT by analyzing its empirical risk, a geometric contraction property induced by the structural bias, and an explicit upper bound on the population risk. Detailed proofs and supporting derivations are provided in Appendix B.

**Definition 4.1** (Empirical Risk). Consider an input image space $\mathcal{X}$, a discrete label space $\mathcal{Y}$, and a distribution $\mathcal{D}$ over $\mathcal{X} \times \mathcal{Y}$. Let $\mathcal{C} = \{1, \ldots, C\}$ be the label index set. Let $\mathcal{D}^{\text{train}} = \{(\mathbf{x}_i, y_i)\}_{i=1}^N$ be a training set drawn i.i.d. from $\mathcal{D}$. Let $\boldsymbol{\ell}_{\mathbf{x}}^{\boldsymbol{\theta}} \in \mathbb{R}^C$ be the logit vector with parameters $\boldsymbol{\theta}$ for input $\mathbf{x}$. Let $p^{\boldsymbol{\theta}}(c \mid \mathbf{x}) = \exp(\ell_c^{\boldsymbol{\theta}}(\mathbf{x})) / \sum_{c' \in \mathcal{C}} \exp(\ell_{c'}^{\boldsymbol{\theta}}(\mathbf{x}))$ be the softmax prediction. Define the empirical risk

$$\widehat{R}_N(\boldsymbol{\theta}) = -\frac{1}{N} \sum_{i=1}^N \log p^{\boldsymbol{\theta}}(y_i \mid \mathbf{x}_i).$$

**Lemma 4.2** (Contraction Induced by Additive Fusion). *Let $d$ be the embedding dimension and let $\mathbb{S}^{d-1} = \{\mathbf{z} \in \mathbb{R}^d : \|\mathbf{z}\|_2 = 1\}$ be the unit sphere. Let $\boldsymbol{\varphi} \in \mathbb{S}^{d-1}$ and $\boldsymbol{\psi} \in \mathbb{S}^{d-1}$ satisfy $\boldsymbol{\varphi} + \boldsymbol{\psi} \neq 0$, which is equivalent to $\langle \boldsymbol{\varphi}, \boldsymbol{\psi} \rangle > -1$. Define the fused unit vector $\boldsymbol{\omega} = (\boldsymbol{\varphi} + \boldsymbol{\psi}) / \|\boldsymbol{\varphi} + \boldsymbol{\psi}\|_2$. Then the fusion map contracts toward $\boldsymbol{\varphi}$ in the sense that*

$$\|\boldsymbol{\omega} - \boldsymbol{\varphi}\|_2^2 \leq \frac{1}{2} \|\boldsymbol{\psi} - \boldsymbol{\varphi}\|_2^2.$$

Lemma 4.2 demonstrates that the structural bias constrains the final representation to lie geometrically closer to the frozen reference than the prompt-only representation. This contraction property provides the theoretical basis for the incremental correction mechanism.

**Lemma 4.3.** *Let $\tau > 0$ be the temperature and define logits $\ell_c^{\boldsymbol{\theta}}(\mathbf{x}) = \tau \langle \mathbf{f}_{\mathbf{x}}^{\text{vis}}, \mathbf{f}_c^{\text{txt}} \rangle$ and reference logits $\ell_c^0(\mathbf{x}) = \tau \langle \mathbf{z}_{\mathbf{x}}^{\text{vis}}, \mathbf{w}_c \rangle$. Define logit perturbation vector $\Delta \boldsymbol{\ell}_{\mathbf{x}}^{\boldsymbol{\theta}} = \boldsymbol{\ell}_{\mathbf{x}}^{\boldsymbol{\theta}} -$*

$\ell_{\mathbf{x}}^0 \in \mathbb{R}^C$. *Define the consistency regularizer* $\mathcal{L}^{\mathrm{con}}(\boldsymbol{\theta}) = \mathcal{L}^{\mathrm{img}}(\boldsymbol{\theta}) + \mathcal{L}^{\mathrm{txt}}(\boldsymbol{\theta})$. *Then for training samples* $\{\mathbf{x}_i\}_{i=1}^N$,

$$\frac{1}{N}\sum_{i=1}^{N}\|\Delta\ell_{\mathbf{x}_i}^{\boldsymbol{\theta}}\|_2^2 \leq 4\tau^2 C\, \mathcal{L}^{\mathrm{con}}(\boldsymbol{\theta}).$$

Lemma 4.3 shows that cosine consistency explicitly bounds the logit perturbation magnitude relative to the frozen reference model, ensuring the feature adaptation is confined within the geometric neighborhood.

**Corollary 4.4.** *Let* $\phi_y(\boldsymbol{\ell}) = \log\sum_{c\in\mathcal{C}}\exp(\ell_c) - \ell_y$ *be the cross-entropy loss for label* $y$ *and logit vector* $\boldsymbol{\ell}$. *Let the population risk be* $R(\boldsymbol{\theta}) = \mathbb{E}^{(\mathbf{x},y)\sim\mathcal{D}}[\phi_y(\boldsymbol{\ell}_{\mathbf{x}}^{\boldsymbol{\theta}})]$ *and let* $\widehat{R}_N(\boldsymbol{\theta}) = \frac{1}{N}\sum_{i=1}^N \phi_{y_i}(\boldsymbol{\ell}_{\mathbf{x}_i}^{\boldsymbol{\theta}})$ *be the empirical risk defined in Definition 4.1. Let* $\varepsilon > 0$ *and define the localized parameter set* $\Omega^\varepsilon = \{\boldsymbol{\theta} : \mathcal{L}^{\mathrm{con}}(\boldsymbol{\theta}) \leq \varepsilon\}$. *Let* $B = \log C + 2\tau$ *and let* $\rho \in (0,1)$ *be the confidence level. With probability at least* $1 - \rho$, *for all* $\boldsymbol{\theta} \in \Omega^\varepsilon$,

$$R(\boldsymbol{\theta}) \leq \widehat{R}_N(\boldsymbol{\theta}) + 2\widehat{\mathfrak{R}}_N(\mathcal{H}^\varepsilon) + 4B\sqrt{\frac{\log(4/\rho)}{2N}}.$$

*Let* $\mathcal{H}^\varepsilon = \{(\mathbf{x},y) \mapsto \phi_y(\boldsymbol{\ell}_{\mathbf{x}}^{\boldsymbol{\theta}}) - \phi_y(\boldsymbol{\ell}_{\mathbf{x}}^0) : \boldsymbol{\theta} \in \Omega^\varepsilon\}$ *be the induced difference class and let* $\widehat{\mathfrak{R}}_N(\mathcal{H}^\varepsilon)$ *be its empirical Rademacher complexity. Assume there exist constants* $\Lambda^{\mathcal{H}} > 0$ *and* $R^{\boldsymbol{\theta}} > 0$ *and a prompt dimension* $D^{\boldsymbol{\theta}}$ *such that*

$$\widehat{\mathfrak{R}}_N(\mathcal{H}^\varepsilon) \leq \frac{24\tau}{\sqrt{N}}\sqrt{2C\,\varepsilon}\sqrt{D^{\boldsymbol{\theta}}\log\!\Big(\frac{3e\Lambda^{\mathcal{H}}R^{\boldsymbol{\theta}}}{2\tau\sqrt{2C\,\varepsilon}}\Big)}.$$

Corollary 4.4 shows that ManiPT can achieve a smaller or equal population risk bound when empirical risks are comparable and the consistency loss is reduced, which alleviates overfitting under limited supervision.

# 5. Experiments

**Baselines and Benchmarks.** To evaluate ManiPT, we conduct experiments on 15 widely used datasets covering general object, fine-grained, scene, texture, and satellite image classification, as well as four ImageNet variants.

We compare ManiPT with state-of-the-art methods including Zero-shot CLIP (Radford et al., 2021), basic prompt tuning approaches such as CoOp (Zhou et al., 2022b) and CoCoOp (Zhou et al., 2022a), deep prompting methods such as MaPLe (Khattak et al., 2023a), regularization strategies including PromptSRC (Khattak et al., 2023b) and Co-Prompt (Roy & Etemad, 2023), and knowledge-enhanced approaches such as TAC (Hao et al., 2025), TAP (Ding et al., 2024), and LLaMP (Zheng et al., 2024) as baselines.

Following standard protocols (Zhou et al., 2022b), we evaluate performance under four settings. In Base-to-Novel Generalization, classes are split into base and novel sets.

Cross-Dataset Transfer evaluates zero-shot transferability from ImageNet to the other ten datasets. Domain Generalization evaluates robustness on ImageNet (Deng et al., 2009) variants. Few-Shot Classification evaluates performance on all datasets under 1, 2, 4, 8, and 16 shot settings. See Appendix A for dataset and implementation details.

**Performance Comparison.**

**Base-to-Novel Generalization.** We present the base-to-novel generalization results across 11 datasets in Table 1. ManiPT consistently outperforms all comparison methods in terms of the average performance. These results indicate that while consistency constraints confine the learned representations within the pretrained geometric neighborhood, the structural bias further guides the model toward transferable directions. This dual mechanism effectively prevents overfitting to shortcuts while leveraging pretrained knowledge, leading to more balanced generalization.

**Cross-Dataset Transfer.** To evaluate the transferability of the learned prompts to unseen distributions, we train the model on ImageNet and directly evaluate it on the other ten datasets. Table 2 reports the cross-dataset evaluation results. ManiPT achieves the highest average accuracy of 68.04% and outperforms CoPrompt at 66.99% and TAC at 66.53%. These results suggest that by enforcing incremental corrections via structural bias, ManiPT ensures the learned representations align with robust, transferable directions rather than dataset-specific shortcuts, thereby facilitating transfer across substantial cross-domain distribution shifts.

**Domain Generalization.** We further assess the robustness of the model against domain shifts using ImageNet variants. As reported in Table 3, ManiPT achieves the best average performance and maintains strong accuracy across all target domains. This behavior validates our insight that anchoring the final representations to the frozen backbone allows the model to filter out domain-specific noise, ensuring that the feature adaptation remains along robust semantic directions.

**Few-Shot Classification.** Figure 3 illustrates average performance across 11 datasets under 1, 2, 4, 8, and 16 shot settings. ManiPT consistently outperforms the baseline methods across all shot settings. Even in the most data-scarce cases such as 1-shot and 2-shot, ManiPT maintains clear performance gains. These results support the view that regulating the feature adaptation via structural bias and consistency constraints alleviates overfitting under limited supervision. See Appendix C.1 for detailed few-shot results.

**Ablation Studies.** Table 4 reports ablation results of ManiPT. Removing cosine consistency leads to a significant drop in novel class performance, validating that confining the learned representations to the geometric neighborhood is necessary to prevent catastrophic drift. Removing the structural bias degrades performance on both base and novel

*Table 1.* Base-to-novel generalization results on 11 datasets. We report accuracy on base and novel classes and their harmonic mean (HM). ManiPT achieves the best average harmonic mean across datasets, indicating more balanced generalization between base and novel classes.

| Method | (a) Average | | | (b) ImageNet | | | (c) Caltech101 | | | (d) OxfordPets | | |
|---|---|---|---|---|---|---|---|---|---|---|---|---|
| | Base | Novel | HM | Base | Novel | HM | Base | Novel | HM | Base | Novel | HM |
| CLIP | 69.34 | 74.22 | 71.70 | 72.43 | 68.14 | 70.22 | 96.84 | 94.00 | 95.40 | 91.17 | 97.26 | 94.12 |
| CoOp | 82.69 | 63.22 | 71.66 | 76.47 | 67.88 | 71.92 | 98.00 | 89.81 | 93.73 | 93.67 | 95.29 | 94.47 |
| CoCoOp | 80.47 | 71.69 | 75.83 | 75.98 | 70.43 | 73.10 | 97.96 | 93.81 | 95.84 | 95.20 | 97.69 | 96.43 |
| MaPLe | 82.28 | 75.14 | 78.55 | 76.66 | 70.54 | 73.47 | 97.74 | 94.36 | 96.02 | 95.43 | 97.76 | 96.58 |
| LLaMP | 85.16 | 77.71 | 81.27 | 77.99 | 71.27 | 74.48 | 98.45 | 95.85 | 97.13 | 96.31 | 97.74 | 97.02 |
| PromptSRC | 84.26 | 76.10 | 79.97 | 77.60 | 70.73 | 74.01 | 98.10 | 94.03 | 96.02 | 95.33 | 97.30 | 96.30 |
| CoPrompt | 84.00 | 77.23 | 80.48 | 77.67 | 71.27 | 74.33 | 98.27 | 94.90 | 96.55 | 95.67 | 98.10 | 96.87 |
| TAC | 85.24 | 77.60 | 81.24 | 78.57 | 71.03 | 74.61 | 98.57 | 95.27 | 96.89 | 95.93 | 98.17 | 97.04 |
| TAP | 84.75 | 77.63 | 81.04 | 77.97 | 70.40 | 73.99 | 98.90 | 95.50 | 97.17 | 95.80 | 97.73 | 96.76 |
| **ManiPT** | **85.82** | **78.67** | **82.09** | **78.67** | **72.01** | **75.20** | **98.95** | **95.94** | **97.42** | **96.52** | 98.15 | **97.33** |

| Method | (e) StanfordCars | | | (f) Flowers102 | | | (g) Food101 | | | (h) FGVCAircraft | | |
|---|---|---|---|---|---|---|---|---|---|---|---|---|
| | Base | Novel | HM | Base | Novel | HM | Base | Novel | HM | Base | Novel | HM |
| CLIP | 63.37 | 74.89 | 68.65 | 72.08 | **77.80** | 74.83 | 90.10 | 91.22 | 90.66 | 27.19 | 36.29 | 31.09 |
| CoOp | 78.12 | 60.40 | 68.13 | 97.60 | 59.67 | 74.06 | 88.33 | 82.26 | 85.19 | 40.44 | 22.30 | 28.75 |
| CoCoOp | 70.49 | 73.59 | 72.01 | 94.87 | 71.75 | 81.71 | 90.70 | 91.29 | 90.99 | 33.41 | 23.71 | 27.74 |
| MaPLe | 72.94 | 74.00 | 73.47 | 95.92 | 72.46 | 82.56 | 90.71 | 92.05 | 91.38 | 37.44 | 35.61 | 36.50 |
| LLaMP | 81.56 | 74.54 | 77.89 | 97.82 | 77.40 | **86.42** | 91.05 | 91.93 | 91.49 | 47.30 | 37.61 | 41.90 |
| PromptSRC | 78.27 | 74.97 | 76.58 | 98.07 | 76.50 | 85.95 | 90.67 | 91.53 | 91.10 | 42.73 | 37.87 | 40.15 |
| CoPrompt | 76.97 | 74.40 | 75.66 | 97.27 | 76.60 | 85.71 | 90.73 | 92.07 | 91.40 | 40.20 | 39.33 | 39.76 |
| TAC | 81.63 | 74.17 | 77.72 | 97.97 | 76.87 | 86.15 | 90.87 | 91.87 | 91.37 | 44.60 | 37.70 | 40.86 |
| TAP | 80.70 | 74.27 | 77.35 | 97.90 | 75.57 | 85.30 | 90.97 | 91.83 | 91.40 | 44.40 | 36.50 | 40.06 |
| **ManiPT** | **82.53** | **75.61** | **78.92** | **98.12** | 76.84 | 86.19 | **91.17** | **92.16** | **91.66** | **48.28** | **39.87** | **43.68** |

| Method | (i) SUN397 | | | (j) DTD | | | (k) EuroSAT | | | (l) UCF101 | | |
|---|---|---|---|---|---|---|---|---|---|---|---|---|
| | Base | Novel | HM | Base | Novel | HM | Base | Novel | HM | Base | Novel | HM |
| CLIP | 69.36 | 75.35 | 72.23 | 53.24 | 59.90 | 56.37 | 56.48 | 64.05 | 60.03 | 70.53 | 77.50 | 73.85 |
| CoOp | 80.60 | 65.89 | 72.51 | 79.44 | 41.18 | 54.24 | 92.19 | 54.74 | 68.69 | 84.69 | 56.05 | 67.46 |
| CoCoOp | 79.74 | 76.86 | 78.27 | 77.01 | 56.00 | 64.85 | 87.49 | 60.04 | 71.21 | 82.33 | 73.45 | 77.64 |
| MaPLe | 80.82 | 78.70 | 79.75 | 80.36 | 59.18 | 68.16 | 94.07 | 73.23 | 82.35 | 83.00 | 78.66 | 80.77 |
| LLaMP | 83.41 | 79.90 | 81.62 | 83.49 | 64.49 | 72.77 | 91.93 | 83.66 | 87.60 | 87.13 | 80.66 | 83.77 |
| PromptSRC | 82.67 | 78.47 | 80.52 | 83.37 | 62.97 | 71.75 | 92.90 | 73.90 | 82.32 | 87.10 | 78.80 | 82.74 |
| CoPrompt | 82.63 | 80.03 | 81.31 | 83.13 | 64.73 | 72.79 | **94.60** | 78.57 | 85.84 | 86.90 | 79.57 | 83.07 |
| TAC | **83.70** | 80.03 | **81.82** | 83.37 | 64.27 | 72.58 | 94.37 | 82.60 | 88.10 | 88.07 | 81.67 | 84.75 |
| TAP | 82.87 | 79.53 | 81.17 | 84.20 | 68.00 | 75.24 | 90.70 | 82.17 | 86.22 | 87.90 | **82.43** | 85.08 |
| **ManiPT** | 83.18 | **80.18** | 81.65 | **84.38** | **68.56** | **75.65** | 94.06 | **83.72** | **88.59** | **88.14** | 82.31 | **85.13** |

*Table 2.* Cross-dataset transfer results from ImageNet to ten downstream datasets. Models are trained on ImageNet and evaluated in a zero-shot manner on each target dataset. ManiPT attains the highest average accuracy over all target datasets.

| | Source | Target | | | | | | | | | | |
|---|---|---|---|---|---|---|---|---|---|---|---|---|
| | ImageNet | Caltech101 | OxfordPets | StanfordCars | Flowers102 | Food101 | Aircraft | SUN397 | DTD | EuroSAT | UCF101 | Average |
| CoOp | 71.51 | 93.70 | 89.14 | 64.51 | 68.71 | 85.30 | 18.47 | 64.15 | 41.92 | 46.39 | 66.55 | 63.88 |
| CoCoOp | 71.02 | 94.43 | 90.14 | 65.32 | 71.88 | 86.06 | 22.94 | 67.36 | 45.73 | 45.37 | 68.21 | 65.74 |
| MaPLe | 70.72 | **95.53** | 90.49 | 65.57 | 72.20 | 86.20 | 24.74 | 67.01 | 46.49 | 48.06 | 68.69 | 66.50 |
| CoPrompt | 70.80 | 94.50 | 90.73 | 65.67 | 72.30 | 86.43 | 24.00 | 67.57 | 47.07 | 51.90 | 69.73 | 66.99 |
| TAC | 72.77 | 94.53 | 90.67 | 65.30 | 72.20 | 85.83 | 23.53 | 67.63 | 47.57 | 48.07 | 70.00 | 66.53 |
| TAP | 72.30 | 94.30 | 90.70 | 65.60 | 70.93 | 86.10 | 24.57 | 68.30 | 50.20 | 46.00 | 68.90 | 66.56 |
| ManiPT | 72.92 | 95.13 | 90.87 | 66.27 | 72.14 | 86.66 | 27.00 | 68.50 | 51.83 | 50.88 | 71.11 | 68.04 |

*Table 3.* Domain generalization results on ImageNet variants. Models trained on ImageNet are tested on distribution-shifted versions such as ImageNet-V2, ImageNet-Sketch, ImageNet-A, and ImageNet-R. ManiPT achieves the best average performance and maintains strong accuracy across all domains.

| | Source | | Target | | | | |
|---|---|---|---|---|---|---|---|
| | ImageNet | | -V2 | -S | -A | -R | Avg. |
| CoOp | 71.51 | | 64.20 | 47.99 | 49.71 | 75.21 | 59.28 |
| CoCoOp | 71.02 | | 64.07 | 48.75 | 50.63 | 76.18 | 59.91 |
| MaPLe | 70.72 | | 64.07 | 49.15 | 50.90 | 76.98 | 60.27 |
| CoPrompt | 70.80 | | 64.25 | 49.43 | 50.50 | 77.51 | 60.42 |
| TAC | 72.77 | | 65.97 | 50.30 | 51.73 | 78.50 | 61.63 |
| ManiPT | **72.92** | | **66.12** | **50.63** | **51.97** | **78.79** | **61.88** |

classes. This confirms that even within the neighborhood, the structural bias is crucial to enforce incremental corrections and guide the adaptation along transferable directions, rather than overfitting to local shortcuts. We also analyze the impact of LLM-based knowledge enrichment. When we substitute the semantic prototypes derived from LLMs with standard templates of the form "a photo of a {class}", the performance decreases, which indicates that constructing stable semantic references with prior knowledge is beneficial. Combining all components yields the best overall

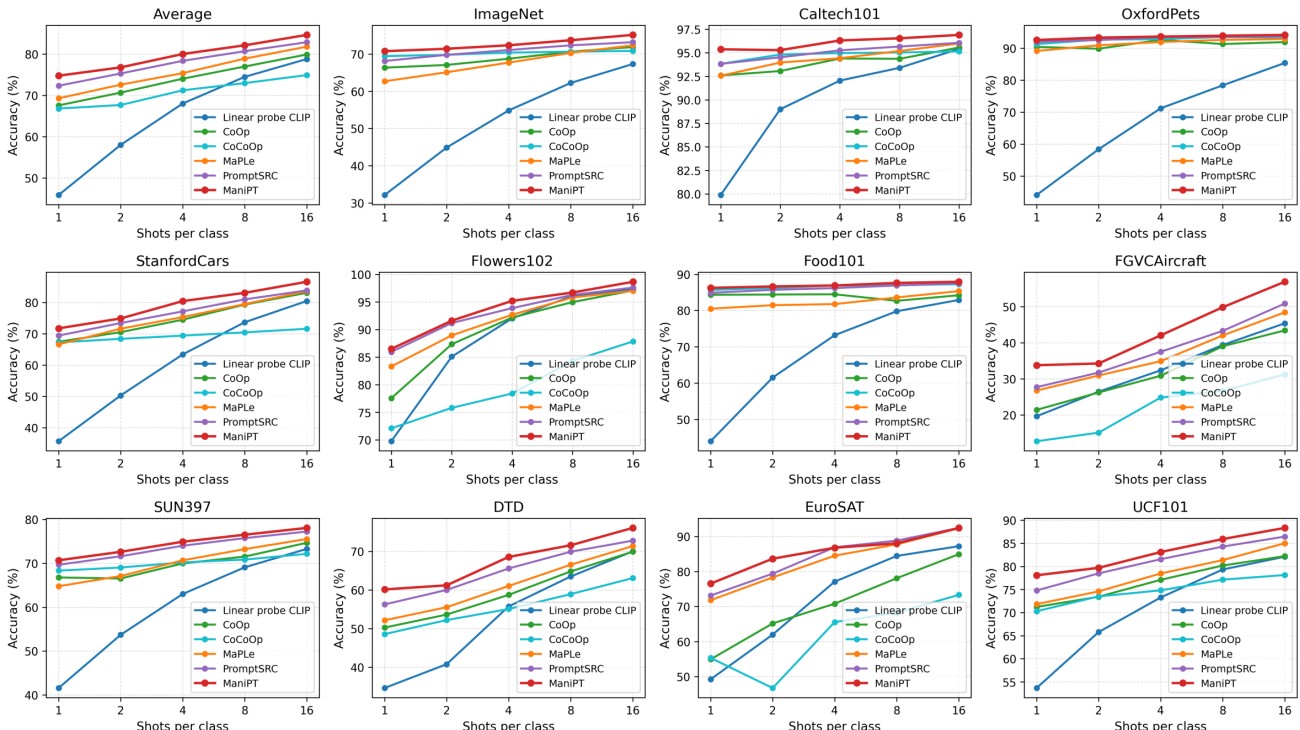

*Figure 3.* Few-shot classification performance averaged over 11 datasets under 1, 2, 4, 8, and 16 shot settings. ManiPT consistently outperforms baseline methods across all shot numbers, with especially clear gains in the 1-shot and 2-shot regimes.

*Table 4.* Ablation study of ManiPT components. We report performance for the baseline, variants with a single component removed, and the full ManiPT model. Each component brings improvements over the baseline, and combining cosine consistency, structural bias, and LLM-based enrichment yields the best overall trade-off.

| Structural Bias | Cos. Consistency | LLM Enrich. | Base | New | HM |
|:---:|:---:|:---:|:---:|:---:|:---:|
| ✗ | ✗ | ✗ | 82.91 | 64.15 | 72.82 |
| ✓ | ✗ | ✗ | 84.85 | 68.32 | 75.71 |
| ✗ | ✓ | ✗ | 82.85 | 76.50 | 79.54 |
| ✓ | ✓ | ✗ | 85.65 | 77.58 | 81.42 |
| ✓ | ✓ | ✓ | **85.82** | **78.67** | **82.09** |

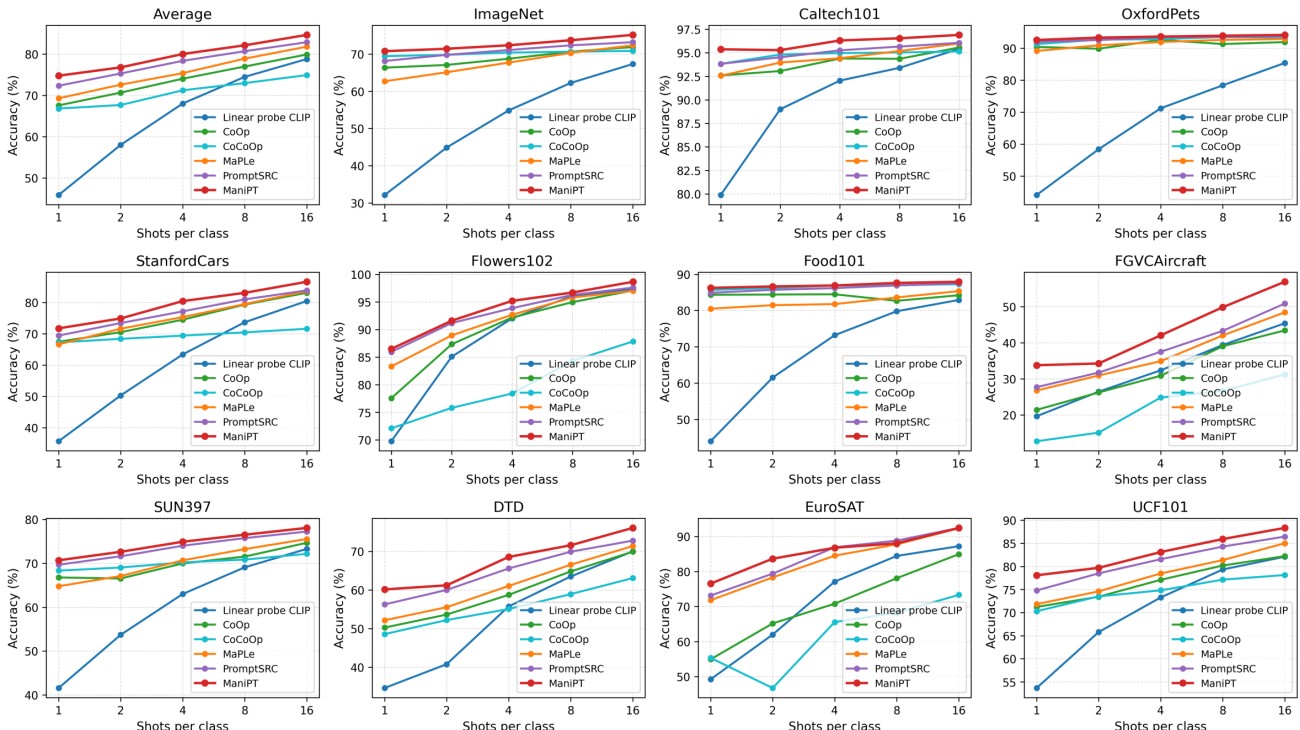

*Figure 4.* Quantitative analysis of manifold drift using PCA. We compare the distance between prompt-adapted features and pretrained features for different methods.

trade-off. See Appendix C.2 for additional ablation results and Appendix C.3 for hyperparameter sensitivity analysis.

**Quantitative Analysis of Manifold Drift.** Since the true pretrained manifold is not directly accessible and the visual modality provides more samples, we follow Eq. (6) to approximate it. Specifically, utilizing the model trained on the

base class training set, we perform the approximation using all images from the novel class test set in each dataset. The reference manifold is approximated by the PCA principal subspace spanned by the top 64 principal components of the pretrained feature point cloud derived from these novel images. We then measure the degree of deviation of the prompt-tuned features from this principal subspace. Figure 4 compares ManiPT with a representative deep prompting baseline across datasets. ManiPT consistently achieves smaller estimated drift and better generalization to novel classes. These observations support the main claim that confining the feature adaptation within the pretrained manifold allows the model to find valid solutions while minimizing the risk of drifting into non-transferable subspaces. See Appendix D.2 for a sensitivity analysis with respect to $d^{\mathrm{pca}}$.

## 6. Conclusion

In this paper, we propose ManiPT to address generalization degradation caused by manifold drift in prompt tuning. ManiPT confines representations within the pretrained neighborhood while enforcing incremental corrections. This dual mechanism biases adaptation toward transferable directions, mitigating overfitting to dataset-specific shortcuts in low-data regimes. Extensive experiments and theoretical analysis validate our framework. By elucidating geometric confinement and directional correction, ManiPT provides a new perspective on preventing low-data overfitting. We discuss the limitations of our framework in E.

## Impact Statement

This paper presents work whose goal is to advance the field of machine learning. There are many potential societal consequences of our work, none of which we feel must be specifically highlighted here.

## Acknowledgements

This work was supported in part by the National Natural Science Foundation of China under Grants 62441232, 62476068, 62306092, 62502115, and 62306091, in part by the Shandong Provincial Natural Science Foundation under Grants ZR2025ZD01, ZR2023QF052, ZR2024QF066, ZR2025QC1516, and in part by the Major Science and Technology Special Project of Inner Mongolia Autonomous Region under Grant 2024DXZD0004.

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

# A. Appendix 1: More Training Information

## A.1. Dataset Information

*Table 5.* Statistics of all 15 datasets used in our experiments, including standard classification benchmarks and ImageNet variants for domain generalization.

| Dataset | Domain / Task | Classes | Train Size | Test Size |
|---|---|---|---|---|
| ***Standard Benchmarks*** | | | | |
| ImageNet | General Objects | 1,000 | 1,281,167 | 50,000 |
| Caltech101 | General Objects | 100 | 4,128 | 2,465 |
| OxfordPets | Fine-grained (Pets) | 37 | 2,944 | 3,669 |
| StanfordCars | Fine-grained (Cars) | 196 | 6,509 | 8,041 |
| Flowers102 | Fine-grained (Flowers) | 102 | 4,093 | 2,463 |
| Food101 | Fine-grained (Food) | 101 | 50,500 | 30,300 |
| FGVCAircraft | Fine-grained (Aircraft) | 100 | 3,334 | 3,333 |
| SUN397 | Scene Recognition | 397 | 15,880 | 19,850 |
| DTD | Texture Classification | 47 | 2,820 | 1,692 |
| EuroSAT | Satellite Imagery | 10 | 13,500 | 8,100 |
| UCF101 | Action Recognition | 101 | 7,639 | 3,783 |
| ***ImageNet Variants (Domain Generalization)*** | | | | |
| ImageNet-V2 | Natural Shift | 1,000 | - | 10,000 |
| ImageNet-Sketch | Style Shift (Sketches) | 1,000 | - | 50,889 |
| ImageNet-A | Natural Adversarial | 200 | - | 7,500 |
| ImageNet-R | Style Shift (Renditions) | 200 | - | 30,000 |

We follow previous work (Zhou et al., 2022b) to set up the benchmark and adopt the same method to split training and test sets. All datasets are publicly available. To evaluate the effectiveness of ManiPT, we employ a total of 15 datasets that cover a wide range of visual tasks and domains. For standard evaluations including base-to-novel generalization, cross-dataset transfer, and few-shot classification, we select 11 datasets that span varying semantic granularities. These include general object recognition datasets ImageNet (Deng et al., 2009) and Caltech101 (Fei-Fei et al., 2004), alongside five fine-grained classification datasets, specifically OxfordPets (Parkhi et al., 2012), StanfordCars (Krause et al., 2013), Flowers102 (Nilsback & Zisserman, 2008), Food101 (Bossard et al., 2014), and FGVCAircraft (Maji et al., 2013) as fine-grained benchmarks. Additionally, we incorporate datasets for distinct modalities, namely SUN397 (Xiao et al., 2010) for scene recognition, UCF101 (Soomro et al., 2012) for action recognition, DTD (Cimpoi et al., 2014) for texture classification, and EuroSAT (Helber et al., 2019) for satellite remote sensing. Furthermore, to assess robustness in the domain generalization setting, we utilize four variants of ImageNet as target domains representing distinct distribution shifts. These consist of ImageNet-V2 (Recht et al., 2019) which contains natural distribution shifts, ImageNet-Sketch (Wang et al., 2019) which tests structural invariance with hand-drawn sketches, ImageNet-A (Hendrycks et al., 2021b) which comprises natural adversarial examples, and ImageNet-R (Hendrycks et al., 2021a) which encompasses artistic renditions. Table 5 summarizes the detailed statistics for all employed datasets.

## A.2. Implementation Details

In all experiments, we utilize the pretrained CLIP ViT-B/16 as the backbone. We adopt a deep prompting strategy (Jia et al., 2022), inserting learnable prompt vectors into every layer of both the visual and text encoders. Specifically, we introduce 16 prompt vectors per layer for each modality. All prompt parameters are initialized using a normal distribution with a standard deviation of 0.02. We employ the Adam optimizer (Kingma & Ba, 2017) with an initial learning rate of $2.5 \times 10^{-3}$ and a weight decay of $5 \times 10^{-4}$. The exponential decay rates for the first and second moment estimates are set to 0.9 and 0.999, respectively, with the numerical stability constant set to $1 \times 10^{-8}$. We apply a constant warm-up for the first epoch with a learning rate of $1 \times 10^{-5}$, followed by a cosine annealing learning rate schedule (Loshchilov & Hutter, 2017) during optimization. To ensure a fair comparison, we follow CoPrompt (Roy & Etemad, 2023) and use GPT-3 (Brown et al., 2020) to generate class descriptions offline for knowledge enrichment. The balance coefficient $\lambda$ in the loss function is fixed at 12 for all datasets and settings. In all experiments, we report the average results over three independent random seeds to mitigate the impact of randomness. For Base-to-Novel generalization and Few-Shot classification tasks, we train the model on the ImageNet dataset for 40 epochs with a batch size of 128. For the other 10 datasets, we train for 50 epochs with a batch size of 32. For Cross-Dataset Transfer and Domain Generalization tasks, we train the model on the ImageNet dataset for 5

epochs. All experiments are conducted using the PyTorch framework (Paszke et al., 2019) on an NVIDIA RTX 5090 GPU.

## A.3. Procedure of ManiPT

We provide the pseudocode of ManiPT in Algorithm 1, which demonstrates the comprehensive and detailed optimization procedures.

---

**Algorithm 1** Pipeline of ManiPT

---

1: **Input:** Few-shot training data $\mathcal{D}^{\text{train}} = \{(\mathbf{x}_i, y_i)\}_{i=1}^{N}$, class set $\mathcal{C}$, hyperparameters $\tau, \lambda$, epoch count $E$, and Pretrained CLIP with text encoder $\mathcal{T}$ and image encoder $\mathcal{V}$.
2: **Output:** Learned text prompts $\mathcal{P}^{\text{txt}}$ and visual prompts $\mathcal{P}^{\text{vis}}$.
3: # Before training: Knowledge Enrichment
4: **for** each class $c \in \mathcal{C}$ **do**
5:     Generate descriptions $A_c$ using LLM
6:     Obtain normalized feature bank $E_c$ using Eq. (8)
7:     Compute semantic prototype $\mathbf{w}_c$
8: **end for**
9: # Begin training
10: Initialize prompt parameters $\mathcal{P}^{\text{txt}}$ and $\mathcal{P}^{\text{vis}}$
11: **for** $e = 1$ to $E$ **do**
12:     **for** each batch $(\mathbf{x}, y)$ in $\mathcal{D}^{\text{train}}$ **do**
13:         Compute frozen features $\mathbf{z}_{\mathbf{x}}^{\text{vis}}$ and $\mathbf{z}_c^{\text{txt}}$
14:         Compute prompt features $\mathbf{h}_{\mathbf{x}}^{\text{vis}}$ and $\mathbf{h}_c^{\text{txt}}$ using Eq. (4) and Eq. (3)
15:         Compute final fused representations $\mathbf{f}_{\mathbf{x}}^{\text{vis}}$ and $\mathbf{f}_c^{\text{txt}}$ using Eq. (11)
16:         Update prompts $\mathcal{P}^{\text{txt}}, \mathcal{P}^{\text{vis}}$ to minimize $\mathcal{L}^{\text{total}}$ using Eq. (14)
17:     **end for**
18: **end for**

---

# B. Appendix 2: Additional Theoretical Justification

Throughout this appendix, let $d \in \mathbb{N}$ be the embedding dimension and denote by $\mathbb{S}^{d-1} = \{\mathbf{z} \in \mathbb{R}^d : \|\mathbf{z}\|_2 = 1\}$ the unit sphere in $\mathbb{R}^d$.

## B.1. Verifiable Anchor Neighborhood

Let $\mathcal{T}$ be the frozen text encoder and recall the class index set $\mathcal{C} = \{1, \ldots, C\}$. Let $T^{\text{all}}$ denote the set of all token sequences admissible as inputs to $\mathcal{T}$. For each class $c \in \mathcal{C}$, let $T_c^{\star} \subset T^{\text{all}}$ be a nonempty finite set of token sequences. Define the realizable class-specific text feature set

$$\mathcal{M}_c^{\star,\text{txt}} = \left\{ \frac{\mathcal{T}(u)}{\|\mathcal{T}(u)\|_2} : u \in T_c^{\star} \right\} \subset \mathbb{S}^{d-1}.$$

Define the global realizable text feature set as

$$\mathcal{M}^{\text{txt}} = \left\{ \frac{\mathcal{T}(u)}{\|\mathcal{T}(u)\|_2} : u \in T^{\text{all}} \right\} \subset \mathbb{S}^{d-1}.$$

For each class $c \in \mathcal{C}$, let the anchor be the unit vector $\mathbf{w}_c \in \mathbb{S}^{d-1}$. Define the discrete projection

$$\tilde{\mathbf{w}}_c \in \arg\min_{z \in \mathcal{M}_c^{\star,\text{txt}}} \|\mathbf{w}_c - z\|_2 \quad \text{and} \quad \hat{d}_c = \|\mathbf{w}_c - \tilde{\mathbf{w}}_c\|_2.$$

Define

$$\zeta^{\max} = \max_{c \in \mathcal{C}} \hat{d}_c \quad \text{and} \quad \varepsilon^{\text{proj}} = \frac{1}{C} \sum_{c \in \mathcal{C}} \|\mathbf{w}_c - \tilde{\mathbf{w}}_c\|_2^2.$$

For a set $M \subset \mathbb{R}^d$, define $\text{dist}(\mathbf{z}, M) = \inf_{\mathbf{w} \in M} \|\mathbf{z} - \mathbf{w}\|_2$. Since $\mathcal{M}_c^{\star,\text{txt}} \subset \mathcal{M}^{\text{txt}}$ for every $c \in \mathcal{C}$, we have

$$\text{dist}(\mathbf{w}_c, \mathcal{M}^{\text{txt}}) \leq \text{dist}(\mathbf{w}_c, \mathcal{M}_c^{\star,\text{txt}}) = \|\mathbf{w}_c - \tilde{\mathbf{w}}_c\|_2 \leq \zeta^{\max}.$$

## B.2. Sphere Identity and Consistency Losses

Let $\mathbb{S}^{d-1} = \{\mathbf{z} \in \mathbb{R}^d : \|\mathbf{z}\|_2 = 1\}$ denote the unit sphere in $\mathbb{R}^d$. For any $\mathbf{b}_1, \mathbf{b}_2 \in \mathbb{S}^{d-1}$,

$$\|\mathbf{b}_1 - \mathbf{b}_2\|_2^2 = 2\big(1 - \langle \mathbf{b}_1, \mathbf{b}_2 \rangle\big).$$

Let $\{(\mathbf{x}_i, y_i)\}_{i=1}^N$ be training samples with $\mathbf{x}_i \in \mathcal{X}$ and $y_i \in \mathcal{C}$. Let $\mathbf{z}_{\mathbf{x}}^{\mathrm{vis}} \in \mathbb{S}^{d-1}$ be the frozen visual feature. Let $\mathbf{f}_{\mathbf{x}}^{\mathrm{vis}} \in \mathbb{S}^{d-1}$ be the final visual feature produced by ManiPT. Let $\mathbf{f}_c^{\mathrm{txt}} \in \mathbb{S}^{d-1}$ be the final text feature for class $c$. Define the visual-side and text-side consistency losses as

$$\mathcal{L}^{\mathrm{img}}(\boldsymbol{\theta}) = \frac{1}{N} \sum_{i=1}^N \Big(1 - \big\langle \mathbf{f}_{\mathbf{x}_i}^{\mathrm{vis}}, \mathbf{z}_{\mathbf{x}_i}^{\mathrm{vis}} \big\rangle\Big) \quad \text{and} \quad \mathcal{L}^{\mathrm{txt}}(\boldsymbol{\theta}) = \frac{1}{C} \sum_{c \in \mathcal{C}} \Big(1 - \big\langle \mathbf{f}_c^{\mathrm{txt}}, \mathbf{w}_c \big\rangle\Big).$$

Define $\mathcal{L}^{\mathrm{con}}(\boldsymbol{\theta}) = \mathcal{L}^{\mathrm{img}}(\boldsymbol{\theta}) + \mathcal{L}^{\mathrm{txt}}(\boldsymbol{\theta})$.

## B.3. Consistency Loss as a Quantitative Control of Representation Shift

Let $\mathbf{z}_i = \mathbf{z}_{\mathbf{x}_i}^{\mathrm{vis}} \in \mathbb{S}^{d-1}$ and $\mathbf{v}_i = \mathbf{f}_{\mathbf{x}_i}^{\mathrm{vis}} \in \mathbb{S}^{d-1}$. Using the sphere identity and averaging over $i$ yields

$$\frac{1}{N} \sum_{i=1}^N \|\mathbf{v}_i - \mathbf{z}_i\|_2^2 = 2\mathcal{L}^{\mathrm{img}}(\boldsymbol{\theta}).$$

Define the frozen visual feature set $\mathcal{M}^{\mathrm{vis}} = \{\mathbf{z}_{\mathbf{x}}^{\mathrm{vis}} : \mathbf{x} \in \mathcal{X}\}$. Since $\mathbf{z}_i \in \mathcal{M}^{\mathrm{vis}}$ for each $i$, we have $\mathrm{dist}(\mathbf{v}_i, \mathcal{M}^{\mathrm{vis}}) \leq \|\mathbf{v}_i - \mathbf{z}_i\|_2$, hence

$$\frac{1}{N} \sum_{i=1}^N \mathrm{dist}\big(\mathbf{f}_{\mathbf{x}_i}^{\mathrm{vis}}, \mathcal{M}^{\mathrm{vis}}\big)^2 \leq 2\mathcal{L}^{\mathrm{img}}(\boldsymbol{\theta}).$$

For the text side, the same identity gives

$$\frac{1}{C} \sum_{c \in \mathcal{C}} \big\|\mathbf{f}_c^{\mathrm{txt}} - \mathbf{w}_c\big\|_2^2 = 2\mathcal{L}^{\mathrm{txt}}(\boldsymbol{\theta}).$$

Using $\|\mathbf{b}_1 + \mathbf{b}_2\|_2^2 \leq 2\|\mathbf{b}_1\|_2^2 + 2\|\mathbf{b}_2\|_2^2$ and the definition of $\varepsilon^{\mathrm{proj}}$, for each $c$ we have $\big\|\mathbf{f}_c^{\mathrm{txt}} - \tilde{\mathbf{w}}_c\big\|_2 \leq \big\|\mathbf{f}_c^{\mathrm{txt}} - \mathbf{w}_c\big\|_2 + \|\mathbf{w}_c - \tilde{\mathbf{w}}_c\|_2$, hence

$$\frac{1}{C} \sum_{c \in \mathcal{C}} \big\|\mathbf{f}_c^{\mathrm{txt}} - \tilde{\mathbf{w}}_c\big\|_2^2 \leq 4\mathcal{L}^{\mathrm{txt}}(\boldsymbol{\theta}) + 2\varepsilon^{\mathrm{proj}}.$$

Since $\tilde{\mathbf{w}}_c \in \mathcal{M}_c^{\star,\mathrm{txt}}$, it follows that

$$\frac{1}{C} \sum_{c \in \mathcal{C}} \mathrm{dist}\big(\mathbf{f}_c^{\mathrm{txt}}, \mathcal{M}_c^{\star,\mathrm{txt}}\big)^2 \leq 4\mathcal{L}^{\mathrm{txt}}(\boldsymbol{\theta}) + 2\varepsilon^{\mathrm{proj}}.$$

## B.4. Geometric Contraction and Stability Induced by the Structural Bias

The structural bias is implemented via normalized fusion between the frozen features and the prompt-branch features. Recall $\boldsymbol{\varphi} \in \mathbb{S}^{d-1}$ and $\boldsymbol{\psi} \in \mathbb{S}^{d-1}$, and define $\boldsymbol{\omega} = (\boldsymbol{\varphi} + \boldsymbol{\psi})/\|\boldsymbol{\varphi} + \boldsymbol{\psi}\|_2$ when $\boldsymbol{\varphi} + \boldsymbol{\psi} \neq 0$. The following lemma shows that this fusion contracts toward the frozen feature.

**Lemma B.1.** *Assume $\boldsymbol{\varphi} \in \mathbb{S}^{d-1}$ and $\boldsymbol{\psi} \in \mathbb{S}^{d-1}$ satisfy $\boldsymbol{\varphi} + \boldsymbol{\psi} \neq 0$, which is equivalent to $\langle \boldsymbol{\varphi}, \boldsymbol{\psi} \rangle > -1$. Define the fused unit vector $\boldsymbol{\omega} = (\boldsymbol{\varphi} + \boldsymbol{\psi})/\|\boldsymbol{\varphi} + \boldsymbol{\psi}\|_2$. Then*

$$\|\boldsymbol{\omega} - \boldsymbol{\varphi}\|_2^2 \leq \frac{1}{2}\|\boldsymbol{\psi} - \boldsymbol{\varphi}\|_2^2.$$

*Proof.* Let $\gamma = \langle \boldsymbol{\varphi}, \boldsymbol{\psi} \rangle \in [-1, 1]$. Since $\boldsymbol{\varphi} + \boldsymbol{\psi} \neq 0$, we have $\gamma > -1$ and

$$\|\boldsymbol{\varphi} + \boldsymbol{\psi}\|_2^2 = \|\boldsymbol{\varphi}\|_2^2 + \|\boldsymbol{\psi}\|_2^2 + 2\langle \boldsymbol{\varphi}, \boldsymbol{\psi} \rangle = 2(1 + \gamma).$$

Moreover,

$$\langle \boldsymbol{\omega}, \boldsymbol{\varphi} \rangle = \frac{\langle \boldsymbol{\varphi} + \boldsymbol{\psi}, \boldsymbol{\varphi} \rangle}{\|\boldsymbol{\varphi} + \boldsymbol{\psi}\|_2} = \frac{1 + \gamma}{\sqrt{2(1 + \gamma)}} = \sqrt{\frac{1 + \gamma}{2}}.$$

Using the sphere identity $\|\mathbf{b}_1 - \mathbf{b}_2\|_2^2 = 2(1 - \langle \mathbf{b}_1, \mathbf{b}_2 \rangle)$ for $\mathbf{b}_1, \mathbf{b}_2 \in \mathbb{S}^{d-1}$ yields

$$\|\boldsymbol{\omega} - \boldsymbol{\varphi}\|_2^2 = 2 \left( 1 - \sqrt{\frac{1 + \gamma}{2}} \right), \qquad \|\boldsymbol{\psi} - \boldsymbol{\varphi}\|_2^2 = 2(1 - \gamma).$$

Therefore, it suffices to show

$$2 \left( 1 - \sqrt{\frac{1 + \gamma}{2}} \right) \leq 1 - \gamma \iff \sqrt{\frac{1 + \gamma}{2}} \geq \frac{1 + \gamma}{2}.$$

Let $q = (1 + \gamma)/2 \in (0, 1]$. Then the inequality becomes $\sqrt{q} \geq q$, which holds for all $q \in [0, 1]$. This completes the proof. $\qquad \square$

To ensure the fusion is well defined and stable, we impose a non-near-opposition condition. Assume there exist constants $\kappa^{\mathrm{vis}} \in (0, 2]$ and $\kappa^{\mathrm{txt}} \in (0, 2]$ such that for every training image $x$ and every class $c$,

$$\langle \mathbf{z}_{\mathbf{x}}^{\mathrm{vis}}, \mathbf{h}_{\mathbf{x}}^{\mathrm{vis}} \rangle \geq -1 + \kappa^{\mathrm{vis}} \quad \text{and} \quad \langle \mathbf{z}_c^{\mathrm{txt}}, \mathbf{h}_c^{\mathrm{txt}} \rangle \geq -1 + \kappa^{\mathrm{txt}},$$

where $\mathbf{h}_{\mathbf{x}}^{\mathrm{vis}} \in \mathbb{S}^{d-1}$ and $\mathbf{h}_c^{\mathrm{txt}} \in \mathbb{S}^{d-1}$ are the prompt-branch features and $\mathbf{z}_c^{\mathrm{txt}} \in \mathbb{S}^{d-1}$ is the frozen text feature. Then $\left\|\mathbf{z}_{\mathbf{x}}^{\mathrm{vis}} + \mathbf{h}_{\mathbf{x}}^{\mathrm{vis}}\right\|_2^2 \geq 2\kappa^{\mathrm{vis}}$ and $\left\|\mathbf{z}_c^{\mathrm{txt}} + \mathbf{h}_c^{\mathrm{txt}}\right\|_2^2 \geq 2\kappa^{\mathrm{txt}}$. Define the fusion map $\Psi(\boldsymbol{\psi}) = (\boldsymbol{\varphi} + \boldsymbol{\psi})/\|\boldsymbol{\varphi} + \boldsymbol{\psi}\|_2$ for a fixed $\boldsymbol{\varphi} \in \mathbb{S}^{d-1}$.

**Empirical verification of the non-near-opposition condition and the role of $\lambda$.** The non-near-opposition assumption excludes the degenerate case in which the normalized fusion becomes ill defined. When $\langle \boldsymbol{\varphi}, \boldsymbol{\psi} \rangle$ approaches $-1$, the norm $\|\boldsymbol{\varphi} + \boldsymbol{\psi}\|_2$ tends to zero, and stability bounds such as Lemma B.2 cease to be applicable. Empirically, this condition is satisfied for the models after training: the prompt-branch feature remains well aligned with the frozen feature, and cosine similarities close to $-1$ are not observed.

Concretely, on the DTD base-to-novel setting, we use the model trained on the base classes and measure the visual-side cosine similarities $\langle \mathbf{h}_{\mathbf{x}}^{\mathrm{vis}}, \mathbf{z}_{\mathbf{x}}^{\mathrm{vis}} \rangle$ and $\langle \mathbf{f}_{\mathbf{x}}^{\mathrm{vis}}, \mathbf{z}_{\mathbf{x}}^{\mathrm{vis}} \rangle$ over all images from the novel classes, where $\mathbf{h}_{\mathbf{x}}^{\mathrm{vis}} \in \mathbb{S}^{d-1}$ denotes the prompt-branch visual feature and $\mathbf{f}_{\mathbf{x}}^{\mathrm{vis}} \in \mathbb{S}^{d-1}$ denotes the final fused feature used for prediction. The summary statistics are

$$\langle \mathbf{h}_{\mathbf{x}}^{\mathrm{vis}}, \mathbf{z}_{\mathbf{x}}^{\mathrm{vis}} \rangle \quad \text{mean } 0.9769, \text{ 5th percentile } 0.9543, \text{ minimum } 0.8993,$$
$$\langle \mathbf{f}_{\mathbf{x}}^{\mathrm{vis}}, \mathbf{z}_{\mathbf{x}}^{\mathrm{vis}} \rangle \quad \text{mean } 0.9942, \text{ 5th percentile } 0.9885, \text{ minimum } 0.9745.$$

The fused feature $\mathbf{f}_{\mathbf{x}}^{\mathrm{vis}}$ is closer to $\mathbf{z}_{\mathbf{x}}^{\mathrm{vis}}$ than $\mathbf{h}_{\mathbf{x}}^{\mathrm{vis}}$, which is consistent with the contraction effect in Lemma B.1.

We also examine whether the consistency weight $\lambda$ prevents representations from rotating away from the frozen branch. Recall that we define the visual consistency loss applied to the final decision feature as

$$\mathcal{L}^{\mathrm{img}}(\boldsymbol{\theta}) = \frac{1}{N} \sum_{i=1}^{N} \left( 1 - \langle \mathbf{f}_{\mathbf{x}_i}^{\mathrm{vis}}, \mathbf{z}_{\mathbf{x}_i}^{\mathrm{vis}} \rangle \right).$$

This implies the average cosine alignment with the frozen feature satisfies

$$\frac{1}{N} \sum_{i=1}^{N} \langle \mathbf{f}_{\mathbf{x}_i}^{\mathrm{vis}}, \mathbf{z}_{\mathbf{x}_i}^{\mathrm{vis}} \rangle = 1 - \mathcal{L}^{\mathrm{img}}(\boldsymbol{\theta}).$$

Therefore, increasing $\lambda$ in $\mathcal{L}^{\mathrm{total}} = \mathcal{L}^{\mathrm{ce}} + \lambda \left( \mathcal{L}^{\mathrm{img}} + \mathcal{L}^{\mathrm{txt}} \right)$ encourages higher alignment of the decision features with the frozen branch on average. Empirically, when we remove the consistency loss terms by taking $\lambda$ to be zero, the mean cosine similarity decreases and the drift metric $\Delta$ reported in Table 6 increases, which indicates a clear tendency to rotate away from the pretrained manifold.

*Table 6.* Full ablation results on the effect of consistency loss weights across all 11 datasets. We compare the default setting ($\lambda = 12$) with the unconstrained setting ($\lambda = 0$). The results consistently show that removing the constraint leads to a significant drop in cosine alignment and a sharp increase in manifold drift $\Delta$, confirming the necessity of geometric regularization.

| Dataset | Mean $\langle \mathbf{f}_{\mathbf{x}}^{\text{vis}}, \mathbf{z}_{\mathbf{x}}^{\text{vis}} \rangle$ | | Drift $\Delta$ | |
|---|---|---|---|---|
| | $\lambda = 12$ | $\lambda = 0$ | $\lambda = 12$ | $\lambda = 0$ |
| ImageNet | 0.9952 | 0.8741 | 0.0045 | 0.1452 |
| Caltech101 | 0.9978 | 0.8950 | -0.0021 | 0.1050 |
| OxfordPets | 0.9965 | 0.8620 | 0.0012 | 0.1523 |
| StanfordCars | 0.9910 | 0.8150 | 0.0078 | 0.2105 |
| Flowers102 | 0.9930 | 0.8245 | -0.0048 | 0.1950 |
| Food101 | 0.9955 | 0.8810 | 0.0023 | 0.1340 |
| FGVCAircraft | 0.9903 | 0.8102 | -0.0016 | 0.2284 |
| SUN397 | 0.9928 | 0.8430 | 0.0039 | 0.1780 |
| DTD | 0.9942 | 0.8284 | -0.0019 | 0.1817 |
| EuroSAT | 0.9916 | 0.7650 | 0.0121 | 0.3284 |
| UCF101 | 0.9925 | 0.7920 | 0.0055 | 0.2480 |

**Lemma B.2.** *Fix $\varphi \in \mathbb{S}^{d-1}$ and let $\Psi(\psi) = (\varphi + \psi)/\|\varphi + \psi\|_2$. If $\|\varphi + \psi_1\|_2 \geq \upsilon$ and $\|\varphi + \psi_2\|_2 \geq \upsilon$ for some constant $\upsilon > 0$, then the fusion map is $2/\upsilon$-Lipschitz in the sense that $\|\Psi(\psi_1) - \Psi(\psi_2)\|_2 \leq \frac{2}{\upsilon}\|\psi_1 - \psi_2\|_2$.*

*Proof.* Let $\mathbf{b}_1 = \varphi + \psi_1$ and $\mathbf{b}_2 = \varphi + \psi_2$. We bound

$$\left\| \frac{\mathbf{b}_1}{\|\mathbf{b}_1\|_2} - \frac{\mathbf{b}_2}{\|\mathbf{b}_2\|_2} \right\|_2 \leq \left\| \frac{\mathbf{b}_1 - \mathbf{b}_2}{\|\mathbf{b}_1\|_2} \right\|_2 + \left\| \mathbf{b}_2 \left( \frac{1}{\|\mathbf{b}_1\|_2} - \frac{1}{\|\mathbf{b}_2\|_2} \right) \right\|_2.$$

Using $\|\|\mathbf{b}_1\|_2 - \|\mathbf{b}_2\|_2\| \leq \|\mathbf{b}_1 - \mathbf{b}_2\|_2$ and $\|\mathbf{b}_2\|_2 \leq \|\mathbf{b}_1\|_2 + \|\mathbf{b}_1 - \mathbf{b}_2\|_2$ yields the bound $\left\| \mathbf{b}_2 \left( \frac{1}{\|\mathbf{b}_1\|_2} - \frac{1}{\|\mathbf{b}_2\|_2} \right) \right\|_2 \leq \frac{\|\mathbf{b}_1 - \mathbf{b}_2\|_2}{\|\mathbf{b}_1\|_2}$. Thus the total is at most $\frac{2\|\mathbf{b}_1 - \mathbf{b}_2\|_2}{\|\mathbf{b}_1\|_2} \leq \frac{2}{\upsilon}\|\mathbf{b}_1 - \mathbf{b}_2\|_2$. Since $\mathbf{b}_1 - \mathbf{b}_2 = \psi_1 - \psi_2$, the claim follows. $\square$

### B.5. Logit Perturbation Magnitude Controlled by the Consistency Loss

Recall the temperature $\tau > 0$. Define logits $\ell_c^{\boldsymbol{\theta}}(\mathbf{x}) = \tau\langle \mathbf{f}_{\mathbf{x}}^{\text{vis}}, \mathbf{f}_c^{\text{txt}} \rangle$ and reference logits $\ell_c^0(\mathbf{x}) = \tau\langle \mathbf{z}_{\mathbf{x}}^{\text{vis}}, \mathbf{w}_c \rangle$. Define the logit perturbation vector $\Delta\ell_{\mathbf{x}}^{\boldsymbol{\theta}} = \ell_{\mathbf{x}}^{\boldsymbol{\theta}} - \ell_{\mathbf{x}}^0 \in \mathbb{R}^C$. Define matrices

$$\mathbf{F}^{\text{txt}} = \begin{bmatrix} \mathbf{f}_1^{\text{txt}} & \cdots & \mathbf{f}_C^{\text{txt}} \end{bmatrix} \in \mathbb{R}^{d \times C} \quad \text{and} \quad \mathbf{W} = \begin{bmatrix} \mathbf{w}_1 & \cdots & \mathbf{w}_C \end{bmatrix} \in \mathbb{R}^{d \times C}.$$

Then for any $\mathbf{x}$,

$$\Delta\ell_{\mathbf{x}}^{\boldsymbol{\theta}} = \tau\left(\mathbf{F}^{\text{txt}}\right)^{\top}\left(\mathbf{f}_{\mathbf{x}}^{\text{vis}} - \mathbf{z}_{\mathbf{x}}^{\text{vis}}\right) + \tau\left(\mathbf{F}^{\text{txt}} - \mathbf{W}\right)^{\top}\mathbf{z}_{\mathbf{x}}^{\text{vis}}.$$

Since $\left\|\mathbf{z}_{\mathbf{x}}^{\text{vis}}\right\|_2 = 1$ and each column of $\mathbf{F}^{\text{txt}}$ has unit norm, we have $\|\mathbf{F}^{\text{txt}}\|_F = \sqrt{C}$ and $\left\|\left(\mathbf{F}^{\text{txt}}\right)^{\top}\right\|_{\text{op}} \leq \|\mathbf{F}^{\text{txt}}\|_F = \sqrt{C}$. Thus

$$\|\Delta\ell_{\mathbf{x}}^{\boldsymbol{\theta}}\|_2 \leq \tau\sqrt{C}\left\|\mathbf{f}_{\mathbf{x}}^{\text{vis}} - \mathbf{z}_{\mathbf{x}}^{\text{vis}}\right\|_2 + \tau\|\mathbf{F}^{\text{txt}} - \mathbf{W}\|_F.$$

Moreover,

$$\|\mathbf{F}^{\text{txt}} - \mathbf{W}\|_F^2 = \sum_{c \in \mathcal{C}}\left\|\mathbf{f}_c^{\text{txt}} - \mathbf{w}_c\right\|_2^2 = 2C\,\mathcal{L}^{\text{txt}}(\boldsymbol{\theta}).$$

Let $\Delta\mathbf{v}_i = \mathbf{f}_{\mathbf{x}_i}^{\text{vis}} - \mathbf{z}_{\mathbf{x}_i}^{\text{vis}}$. Using $\|\mathbf{b}_1 + \mathbf{b}_2\|_2^2 \leq 2\|\mathbf{b}_1\|_2^2 + 2\|\mathbf{b}_2\|_2^2$ yields

$$\|\Delta\ell_{\mathbf{x}_i}^{\boldsymbol{\theta}}\|_2^2 \leq 2\tau^2 C\,\|\Delta\mathbf{v}_i\|_2^2 + 4\tau^2 C\,\mathcal{L}^{\text{txt}}(\boldsymbol{\theta}).$$

Averaging over $i$ and using $\frac{1}{N}\sum_{i=1}^N \|\Delta\mathbf{v}_i\|_2^2 = 2\mathcal{L}^{\text{img}}(\boldsymbol{\theta})$ gives

$$\frac{1}{N}\sum_{i=1}^N \|\Delta\ell_{\mathbf{x}_i}^{\boldsymbol{\theta}}\|_2^2 \leq 4\tau^2 C\,\mathcal{L}^{\text{con}}(\boldsymbol{\theta}).$$

## B.6. Lipschitz and Boundedness Properties of the Cross-Entropy Loss

Recall $\phi_y(\boldsymbol{\ell}) = \log \sum_{c \in \mathcal{C}} \exp(\ell_c) - \ell_y$ for $\boldsymbol{\ell} \in \mathbb{R}^C$. Let $\mathbf{p} = \mathrm{softmax}(\boldsymbol{\ell})$ and let $\mathbf{e}_y$ be the one hot vector. Then $\nabla_{\boldsymbol{\ell}} \phi_y(\boldsymbol{\ell}) = \mathbf{p} - \mathbf{e}_y$ and $\|\mathbf{p} - \mathbf{e}_y\|_2 \le \sqrt{2}$, hence for any $\boldsymbol{\ell}, \boldsymbol{\ell}' \in \mathbb{R}^C$,

$$|\phi_y(\boldsymbol{\ell}) - \phi_y(\boldsymbol{\ell}')| \le \sqrt{2}\, \|\boldsymbol{\ell} - \boldsymbol{\ell}'\|_2.$$

If $\boldsymbol{\ell} \in [-\tau, \tau]^C$, then $\log \sum_j \exp(\ell_j) \le \log C + \tau$ and $-\ell_y \le \tau$, hence

$$0 \le \phi_y(\boldsymbol{\ell}) \le \log C + 2\tau.$$

Recall $B = \log C + 2\tau$.

## B.7. Localized Complexity and Generalization Bounds

Define the population risk $R(\boldsymbol{\theta}) = \mathbb{E}^{(\mathbf{x},y) \sim \mathcal{D}}[\phi_y(\boldsymbol{\ell}_{\mathbf{x}}^{\boldsymbol{\theta}})]$ and the empirical risk $\widehat{R}_N(\boldsymbol{\theta}) = \frac{1}{N} \sum_{i=1}^N \phi_{y_i}(\boldsymbol{\ell}_{\mathbf{x}_i}^{\boldsymbol{\theta}})$. Let $\boldsymbol{\theta} \in \mathbb{R}^{D^{\boldsymbol{\theta}}}$ denote the vector of prompt parameters. Fix $R^{\boldsymbol{\theta}} > 0$ and define $\Omega^{\mathrm{base}} = \{\boldsymbol{\theta} : \|\boldsymbol{\theta}\|_2 \le R^{\boldsymbol{\theta}}\}$. For $\varepsilon > 0$, define $\Omega^\varepsilon = \{\boldsymbol{\theta} \in \Omega^{\mathrm{base}} : \mathcal{L}^{\mathrm{con}}(\boldsymbol{\theta}) \le \varepsilon\}$. Define the difference class $\mathcal{H}^\varepsilon = \{(\mathbf{x},y) \mapsto \phi_y(\boldsymbol{\ell}_{\mathbf{x}}^{\boldsymbol{\theta}}) - \phi_y(\boldsymbol{\ell}_{\mathbf{x}}^0) : \boldsymbol{\theta} \in \Omega^\varepsilon\}$. For $\boldsymbol{\theta} \in \Omega^\varepsilon$ and each sample point define $h_i(\boldsymbol{\theta}) = \phi_{y_i}(\boldsymbol{\ell}_{\mathbf{x}_i}^{\boldsymbol{\theta}}) - \phi_{y_i}(\boldsymbol{\ell}_{\mathbf{x}_i}^0)$. Using the Lipschitz bound for $\phi_y$ and the logit perturbation bound yields

$$\frac{1}{N} \sum_{i=1}^N h_i(\boldsymbol{\theta})^2 \le 2 \cdot \frac{1}{N} \sum_{i=1}^N \|\Delta \boldsymbol{\ell}_{\mathbf{x}_i}^{\boldsymbol{\theta}}\|_2^2 \le 8\tau^2 C\, \varepsilon.$$

Define $J^{\mathrm{emp}}(h) = \left(\frac{1}{N} \sum_{i=1}^N h(\mathbf{x}_i, y_i)^2\right)^{1/2}$ and define $\chi(\varepsilon) = 2\tau\sqrt{2C\,\varepsilon}$. Then every $h \in \mathcal{H}^\varepsilon$ satisfies $J^{\mathrm{emp}}(h) \le \chi(\varepsilon)$.

Assume there exists a constant $\Lambda^{\mathcal{H}} > 0$ such that for any $\boldsymbol{\theta}_1, \boldsymbol{\theta}_2 \in \Omega^{\mathrm{base}}$,

$$\left(\frac{1}{N} \sum_{i=1}^N \big(h_i(\boldsymbol{\theta}_1) - h_i(\boldsymbol{\theta}_2)\big)^2\right)^{1/2} \le \Lambda^{\mathcal{H}} \|\boldsymbol{\theta}_1 - \boldsymbol{\theta}_2\|_2.$$

Then for any $\eta \in (0, \chi(\varepsilon)]$,

$$\mathcal{N}\big(\eta, \mathcal{H}^\varepsilon, J^{\mathrm{emp}}\big) \le \left(\frac{3\Lambda^{\mathcal{H}} R^{\boldsymbol{\theta}}}{\eta}\right)^{D^{\boldsymbol{\theta}}}.$$

By Dudley entropy integral and standard covering-number bounds (Bartlett & Mendelson, 2002; Bartlett et al., 2005),

$$\widehat{\mathfrak{R}}_N\big(\mathcal{H}^\varepsilon\big) \le \frac{12}{\sqrt{N}} \int_0^{\chi(\varepsilon)} \sqrt{\log \mathcal{N}\big(\eta, \mathcal{H}^\varepsilon, J^{\mathrm{emp}}\big)}\, d\eta \le \frac{12\,\chi(\varepsilon)}{\sqrt{N}} \sqrt{D^{\boldsymbol{\theta}} \log\left(\frac{3e\Lambda^{\mathcal{H}} R^{\boldsymbol{\theta}}}{\chi(\varepsilon)}\right)}.$$

Substituting $\chi(\varepsilon) = 2\tau\sqrt{2C\,\varepsilon}$ yields

$$\widehat{\mathfrak{R}}_N\big(\mathcal{H}^\varepsilon\big) \le \frac{24\tau}{\sqrt{N}} \sqrt{2C\,\varepsilon} \sqrt{D^{\boldsymbol{\theta}} \log\left(\frac{3e\Lambda^{\mathcal{H}} R^{\boldsymbol{\theta}}}{2\tau\sqrt{2C\,\varepsilon}}\right)}.$$

**Theorem B.3.** *Fix the confidence level $\rho \in (0, 1)$. With probability at least $1 - \rho$, for all $\boldsymbol{\theta} \in \Omega^\varepsilon$,*

$$R(\boldsymbol{\theta}) \le \widehat{R}_N(\boldsymbol{\theta}) + 2\widehat{\mathfrak{R}}_N\big(\mathcal{H}^\varepsilon\big) + 4B\sqrt{\frac{\log(4/\rho)}{2N}}.$$

## B.8. Adaptive Bound via Peeling

Let $H = \lceil \log_2 N \rceil$ and set $\varepsilon_\nu = 2^{-\nu}$ for $\nu = 0, 1, \ldots, H$. Define the peeling layers $\Omega^\nu = \{\boldsymbol{\theta} \in \Omega^{\mathrm{base}} : \varepsilon_{\nu+1} < \mathcal{L}^{\mathrm{con}}(\boldsymbol{\theta}) \le \varepsilon_\nu\}$. Applying Theorem B.3 to each $\Omega^{\varepsilon_\nu}$ with confidence level $\rho/(H+1)$ and taking a union bound over $\nu$ yields the following result. With probability at least $1 - \rho$, for all $\boldsymbol{\theta} \in \Omega^{\mathrm{base}}$ satisfying $\mathcal{L}^{\mathrm{con}}(\boldsymbol{\theta}) \le 1$,

$$R(\boldsymbol{\theta}) \le \widehat{R}_N(\boldsymbol{\theta}) + \frac{24\tau}{\sqrt{N}} \sqrt{4C\, \mathcal{L}^{\mathrm{con}}(\boldsymbol{\theta})} \sqrt{D^{\boldsymbol{\theta}} \log\left(\frac{3e\Lambda^{\mathcal{H}} R^{\boldsymbol{\theta}}}{2\tau\sqrt{4C\, \mathcal{L}^{\mathrm{con}}(\boldsymbol{\theta})}}\right)} + 4B\sqrt{\frac{\log\left(4(H+1)/\rho\right)}{2N}}.$$

This bound shows that the leading term scales with $\sqrt{\mathcal{L}^{\mathrm{con}}(\boldsymbol{\theta})/N}$ up to logarithmic factors.

*Table 7.* Comparison of ManiPT performance with various methods for each dataset in few-shot setting. Best and second-best results are highlighted in bold and underline, respectively.

| Dataset | Method | 1 shot | 2 shots | 4 shots | 8 shots | 16 shots |
|---|---|---|---|---|---|---|
| Average | Linear probe CLIP | 45.83 | 57.98 | 68.01 | 74.47 | 78.79 |
| | CoOp | 67.56 | 70.65 | 74.02 | 76.98 | 79.89 |
| | CoCoOp | 66.79 | 67.65 | 71.21 | 72.96 | 74.90 |
| | MaPLe | 69.27 | 72.58 | 75.37 | 78.89 | 81.79 |
| | PromptSRC | 72.32 | 75.29 | 78.35 | 80.69 | 82.87 |
| | ManiPT | **74.75** | **76.77** | **80.01** | **82.12** | **84.64** |
| ImageNet | Linear probe CLIP | 32.13 | 44.88 | 54.85 | 62.23 | 67.31 |
| | CoOp | 66.33 | 67.07 | 68.73 | 70.63 | 71.87 |
| | CoCoOp | 69.43 | 69.78 | 70.39 | 70.63 | 70.83 |
| | MaPLe | 62.67 | 65.10 | 67.70 | 70.30 | 72.33 |
| | PromptSRC | 68.13 | 69.77 | 71.07 | 72.33 | 73.17 |
| | ManiPT | **70.76** | **71.41** | **72.35** | **73.70** | **75.13** |
| Caltech101 | Linear probe CLIP | 79.88 | 89.01 | 92.05 | 93.41 | 95.43 |
| | CoOp | 92.60 | 93.07 | 94.40 | 94.37 | 95.57 |
| | CoCoOp | 93.83 | 94.82 | 94.98 | 95.04 | 95.16 |
| | MaPLe | 92.57 | 93.97 | 94.43 | 95.20 | 96.00 |
| | PromptSRC | 93.83 | 94.53 | 95.27 | 95.67 | 96.07 |
| | ManiPT | **95.38** | **95.29** | **96.32** | **96.55** | **96.91** |
| OxfordPets | Linear probe CLIP | 44.06 | 58.37 | 71.17 | 78.36 | 85.34 |
| | CoOp | 90.37 | 89.80 | 92.57 | 91.27 | 91.87 |
| | CoCoOp | 91.27 | 92.64 | 92.81 | 93.45 | 93.34 |
| | MaPLe | 89.10 | 90.87 | 91.90 | 92.57 | 92.83 |
| | PromptSRC | 92.00 | 92.50 | 93.43 | 93.50 | 93.67 |
| | ManiPT | **92.53** | **93.27** | **93.60** | **93.87** | **94.12** |
| StanfordCars | Linear probe CLIP | 35.66 | 50.28 | 63.38 | 73.67 | 80.44 |
| | CoOp | 67.43 | 70.50 | 74.47 | 79.30 | 83.07 |
| | CoCoOp | 67.22 | 68.37 | 69.39 | 70.44 | 71.57 |
| | MaPLe | 66.60 | 71.60 | 75.30 | 79.47 | 83.57 |
| | PromptSRC | 69.40 | 73.40 | 77.13 | 80.97 | 83.83 |
| | ManiPT | **71.72** | **74.87** | **80.42** | **83.05** | **86.59** |
| Flowers102 | Linear probe CLIP | 69.74 | 85.07 | 92.02 | 96.10 | 97.37 |
| | CoOp | 77.53 | 87.33 | 92.17 | 94.97 | 97.07 |
| | CoCoOp | 72.08 | 75.79 | 78.40 | 84.30 | 87.84 |
| | MaPLe | 83.30 | 88.93 | 92.67 | 95.80 | 97.00 |
| | PromptSRC | 85.93 | 91.17 | 93.87 | 96.27 | 97.60 |
| | ManiPT | **86.48** | **91.60** | **95.19** | **96.70** | **98.65** |
| Food101 | Linear probe CLIP | 43.96 | 61.51 | 73.19 | 79.79 | 82.90 |
| | CoOp | 84.33 | 84.40 | 84.47 | 82.67 | 84.20 |
| | CoCoOp | 85.65 | 86.22 | **86.88** | 86.97 | 87.25 |
| | MaPLe | 80.50 | 81.47 | 81.77 | 83.60 | 85.33 |
| | PromptSRC | 84.87 | 85.70 | 86.17 | 86.90 | 87.50 |
| | ManiPT | **86.24** | **86.63** | 86.90 | **87.60** | **87.92** |
| FGVCAircraft | Linear probe CLIP | 19.61 | 26.41 | 32.33 | 39.35 | 45.36 |
| | CoOp | 21.37 | 26.20 | 30.83 | 39.00 | 43.40 |
| | CoCoOp | 12.68 | 15.06 | 24.79 | 26.61 | 31.21 |
| | MaPLe | 26.73 | 30.90 | 34.87 | 42.00 | 48.40 |
| | PromptSRC | 27.67 | 31.70 | 37.47 | 43.27 | 50.83 |
| | ManiPT | **33.75** | **34.23** | **42.03** | **49.82** | **56.88** |
| SUN397 | Linear probe CLIP | 41.58 | 53.70 | 63.00 | 69.08 | 73.28 |
| | CoOp | 66.77 | 66.53 | 69.97 | 71.53 | 74.67 |
| | CoCoOp | 68.33 | 69.03 | 70.21 | 70.84 | 72.15 |
| | MaPLe | 64.77 | 67.10 | 70.67 | 73.23 | 75.53 |
| | PromptSRC | 69.67 | 71.60 | 74.00 | 75.73 | 77.23 |
| | ManiPT | **70.68** | **72.61** | **74.91** | **76.51** | **78.06** |
| DTD | Linear probe CLIP | 34.59 | 40.76 | 55.71 | 63.46 | 69.96 |
| | CoOp | 50.23 | 53.60 | 58.70 | 64.77 | 69.87 |
| | CoCoOp | 48.54 | 52.17 | 55.04 | 58.89 | 63.04 |
| | MaPLe | 52.13 | 55.50 | 61.00 | 66.50 | 71.33 |
| | PromptSRC | 56.23 | 59.97 | 65.53 | 69.87 | 72.73 |
| | ManiPT | **60.11** | **61.17** | **68.52** | **71.54** | **76.03** |
| EuroSAT | Linear probe CLIP | 49.23 | 61.98 | 77.09 | 84.43 | 87.21 |
| | CoOp | 54.93 | 65.17 | 70.80 | 78.07 | 84.93 |
| | CoCoOp | 55.33 | 46.74 | 65.56 | 68.21 | 73.32 |
| | MaPLe | 71.80 | 78.30 | 84.50 | 87.73 | 92.33 |
| | PromptSRC | 73.13 | 79.37 | **86.90** | **88.80** | **92.43** |
| | ManiPT | **76.54** | **83.64** | 86.77 | 88.02 | 92.40 |
| UCF101 | Linear probe CLIP | 53.66 | 65.78 | 73.28 | 79.34 | 82.11 |
| | CoOp | 71.23 | 73.43 | 77.10 | 80.20 | 82.23 |
| | CoCoOp | 70.30 | 73.51 | 74.82 | 77.14 | 78.14 |
| | MaPLe | 71.83 | 74.60 | 78.47 | 81.37 | 85.03 |
| | PromptSRC | 74.80 | 78.50 | 81.57 | 84.30 | 86.47 |
| | ManiPT | **78.09** | **79.70** | **83.12** | **85.93** | **88.35** |

# C. Appendix 3: More Experimental Results

## C.1. Few-shot experiments

Table 7 reports the numerical results for few-shot classification across all 11 datasets under 1, 2, 4, 8, and 16 shot settings. The results show that ManiPT consistently achieves higher average accuracy than the compared methods across all shot counts. In the 1-shot regime, ManiPT attains the highest average accuracy and exceeds methods such as PromptSRC and MaPLe. As the number of shots increases, ManiPT continues to maintain the lead. On fine-grained datasets such as FGVCAircraft and on texture datasets such as DTD, ManiPT achieves clear gains over baseline methods, which indicates that these datasets are more sensitive to overfitting. These observations support our core insight: while the consistency constraints prevent the feature adaptation from drifting into invalid regions under extreme data scarcity (e.g., 1-shot), the structural bias is essential to guide the limited updates along transferable directions, ensuring that the model learns robust features rather than overfitting to noise.

## C.2. Ablation Studies

### C.2.1. STRUCTURAL BIAS STRATEGY

*Table 8.* Comparison of different structural bias strategies. Our additive structure ensures that the adapted features are geometrically anchored by the pretrained manifold, achieving the best trade-off between base retention and novel generalization.

| Strategy | Base | New | HM |
|---|---|---|---|
| Direct Tuning | 82.85 | 76.50 | 79.54 |
| Gate | 85.35 | 78.40 | 81.73 |
| Additive | **85.82** | **78.67** | **82.09** |

In Table 8, we compare three structural bias strategies. Direct Tuning performs classification using only the prompt-branch features (visual and text), i.e., $\mathbf{f}^{\text{vis,prompt}}$ and $\mathbf{f}^{\text{txt,prompt}}$, and removes the fusion with the frozen features $\mathbf{f}^{\text{vis,clip}}$ and $\mathbf{f}^{\text{txt,clip}}$, which is the standard practice in CoOp and MaPLe. Gate introduces a learnable coefficient $\alpha$ to interpolate between the frozen and prompt features for each modality, e.g., $\alpha \mathbf{f}^{\text{vis,clip}} + (1 - \alpha)\mathbf{f}^{\text{vis,prompt}}$ on the visual side (and analogously for text), to balance pretrained retention and task adaptation. In the few-shot regime, the learnable gate can overfit by pushing $\alpha$ toward an extreme, discarding the frozen branch and weakening the anchoring effect. Our additive strategy employs equal-weight fusion with re-normalization, which physically enforces incremental corrections. By treating the learned prompt as a perturbation rather than a replacement, the structural bias ensures the final decision features remain geometrically anchored to the frozen representations. This mechanism explicitly biases the feature adaptation toward transferable directions inherent in the pretrained backbone, achieving the best trade-off between base accuracy and novel generalization, as reflected by the highest harmonic mean.

### C.2.2. TEXT ANCHOR SOURCE

*Table 9.* Ablation study on different text anchor sources. While handcrafted templates offer a marginal improvement over "a photo of a {class}" template, LLM descriptions provide rich semantic knowledge, significantly boosting generalization on novel classes.

| Anchor Source | Base | New | HM |
|---|---|---|---|
| a photo of a {class} | 85.65 | 77.58 | 81.42 |
| Handcrafted Templates | 85.70 | 77.80 | 81.56 |
| LLM Descriptions | **85.82** | **78.67** | **82.09** |

Table 9 presents the ablation study on different text anchor sources. Using an ensemble of handcrafted templates offers only a marginal improvement over the single template "a photo of a {class}". This suggests that simply increasing the quantity of templates does not guarantee a better semantic prototype. In contrast, utilizing LLM descriptions yields significant improvements. This indicates that leveraging rich semantic knowledge from LLMs constructs a robust geometric center for the textual consistency constraint, which effectively defines a more accurate validity neighborhood for the textual features, thereby enhancing generalization.

*Table 10.* Ablation study on the complementary roles of visual and textual consistency constraints. The results show that $\mathcal{L}^{\text{img}}$ and $\mathcal{L}^{\text{txt}}$ are complementary. While each single constraint improves over the baseline, combining them yields the best performance across all metrics, confirming their synergistic effect.

| $\mathcal{L}^{\text{img}}$ | $\mathcal{L}^{\text{txt}}$ | Base | New | HM |
|---|---|---|---|---|
| × | × | 84.85 | 68.32 | 75.71 |
| ✓ | × | 85.45 | 73.15 | 78.82 |
| × | ✓ | 85.15 | 74.80 | 79.64 |
| ✓ | ✓ | **85.82** | **78.67** | **82.09** |

### C.2.3. COMPLEMENTARITY OF VISUAL AND TEXTUAL CONSISTENCY

Table 10 reports the individual contributions of the visual and textual consistency constraints. The results indicate that the two constraints complement each other. Applying visual consistency alone stabilizes the visual manifold and improves performance on base classes by reducing feature distortion during adaptation. Applying textual consistency alone provides semantic guidance that is more effective for generalizing to novel classes. The full model that employs both constraints achieves the best performance across all metrics. This pattern shows that the visual constraint limits geometric drift in the embedding space, while the textual constraint anchors the semantics. Collectively, they confine the feature adaptation within the robust geometric structure defined by the pretrained multimodal alignment.

### C.2.4. COMPARISON WITH STANDARD REGULARIZATION

*Table 11.* Comparison with standard regularization methods. L1 and L2 denote constraining the distance between prompt-adapted features and pretrained features via L1 and L2 distances, respectively.

| Regularization Method | Base | New | HM |
|---|---|---|---|
| No Constraint | 84.85 | 68.32 | 75.71 |
| L1 | 83.50 | 72.45 | 77.58 |
| L2 | 78.95 | 76.10 | 77.50 |
| Cosine | **85.82** | **78.67** | **82.09** |

Table 11 compares ManiPT with standard regularization techniques. The results show that Euclidean distance penalties (L1 and L2) lead to a noticeable drop in base class performance. In CLIP, classification is driven by cosine similarity between normalized features, so predictions depend on feature direction rather than magnitude. Applying Euclidean penalties in the non-normalized space introduces task-irrelevant magnitude gradients that can interfere with directional alignment. Cosine consistency directly regularizes angular alignment, which is equivalent to confining the features within a hyperspherical neighborhood of the pretrained manifold. This geometric constraint is necessary to prevent the feature adaptation from deviating into orthogonal shortcut subspaces, which Euclidean penalties fail to address effectively in the normalized feature space.

### C.2.5. DISENTANGLING STRUCTURAL BIAS COMPONENTS

*Table 12.* Ablation study on the components of the structural bias.

| Variant | Base | New | HM |
|---|---|---|---|
| ManiPT (Full) | **85.82** | **78.67** | **82.09** |
| w/o Frozen Text ($\mathbf{f}^{\text{txt,clip}}$) | 83.45 | 77.25 | 80.23 |
| w/o Frozen Visual ($\mathbf{f}^{\text{vis,clip}}$) | 85.40 | 78.25 | 81.67 |

Table 12 analyzes the impact of explicitly incorporating frozen CLIP features into the final representation. We observe a distinct asymmetry between modalities. On the textual side, removing the frozen text feature leads to a significant degradation, particularly in Base accuracy. This confirms that prompt-learned text features alone are prone to optimization instability, and the explicit inclusion of the frozen prototype acts as the necessary basis for incremental corrections, ensuring the final representation remains grounded. In contrast, removing the frozen visual feature results in only a marginal drop. This suggests that our visual consistency constraint is highly efficient, successfully restricting the prompt-adapted visual features to the pretrained manifold.

### C.2.6. CONSTRAINT TARGET: PROMPT VS. FINAL FEATURES

*Table 13.* Ablation study on the target of consistency constraints. We compare applying the constraints to the final fused representations versus applying them only to the intermediate prompt features. The results show that constraining the final representation yields better performance.

| Constraint Target | Base | New | HM |
|---|---|---|---|
| Prompt Features (prompt-branch $\mathbf{f}^{\text{vis,prompt}}$, $\mathbf{f}^{\text{txt,prompt}}$) | 84.65 | 77.45 | 80.89 |
| Final Features (fused $\mathbf{f}^{\text{vis,final}}$, $\mathbf{f}^{\text{txt,final}}$) | **85.82** | **78.67** | **82.09** |

Table 13 investigates whether the consistency constraints should be applied to the final fused representations or the intermediate prompt features. The results demonstrate that constraining the final features achieves superior performance. We attribute this to the alignment between the regularization objective and the decision boundary. Since the classification logits are computed using $\mathbf{f}^{\text{vis,final}}$ and $\mathbf{f}^{\text{txt,final}}$, applying consistency constraints directly to these final features ensures that the geometric properties of the actual decision variables are preserved. In contrast, constraining only the prompt-branch features $\mathbf{f}^{\text{vis,prompt}}$ and $\mathbf{f}^{\text{txt,prompt}}$ creates an optimization misalignment: it restricts the direction of the learnable update rather than the final output. This overly restricts the model's plasticity, preventing it from making necessary incremental corrections. By constraining the final fused features, we ensure that the resulting feature adaptation remains valid within the manifold neighborhood while allowing the prompt components sufficient freedom to adjust along transferable directions.

### C.2.7. INITIALIZATION ROBUSTNESS

*Table 14.* Ablation study on initialization strategies. We compare the Normal initialization with Kaiming Uniform initialization. The results indicate that ManiPT is relatively robust to initialization strategies, though Normal initialization yields slightly better performance by avoiding excessive initial variance.

| Initialization | Base | New | HM |
|---|---|---|---|
| Kaiming Uniform | 85.45 | 78.10 | 81.61 |
| Normal (std=0.02) | **85.82** | **78.67** | **82.09** |

Table 14 compares the performance of different initialization strategies. The results show that ManiPT is relatively robust to the choice of initialization, with the harmonic mean dropping only slightly from 82.09% to 81.61% when switching to Kaiming Uniform. However, Normal initialization consistently yields better results. This is likely because Kaiming initialization is designed for training deep networks from scratch and introduces a larger initial variance. In the context of prompt tuning with a frozen backbone, this aggressive initialization can cause the feature adaptation to diverge early into dataset-specific local minima, leading to potential overfitting to base classes and slightly degrading the transferability to novel classes.

### C.2.8. MODALITY-SPECIFIC DEEP PROMPTING

*Table 15.* Ablation study on modality-specific deep prompting. We compare our dual-branch design with single-branch deep prompting variants namely Visual-only and Textual-only. Vis-only and Txt-only denote applying deep prompting, consistency constraints, and structural bias exclusively to the image or text encoder respectively.

| Method | Base | New | HM |
|---|---|---|---|
| Txt-only Deep Prompting | 85.45 | 69.40 | 76.60 |
| Vis-only Deep Prompting | 84.20 | 74.85 | 79.25 |
| ManiPT | 85.82 | 78.67 | 82.09 |

Table 15 investigates the effectiveness of dual-branch deep prompting by comparing it with variants where deep prompting, consistency constraints, and structural bias are applied to a single modality. When restricted to the text encoder, the model exhibits signs of overfitting as it maintains high accuracy on base classes but the generalization to novel classes lags behind the full model. This suggests that adapting only the text encoder tends to optimize decision boundaries for seen classes without sufficiently aligning them with the generalizable visual manifold. Conversely, adapting only the image encoder yields better generalization than the text-only variant reflecting the inherent transferability of visual features yet it still falls short of the performance achieved by the dual-branch design. By coordinating adaptation across both modalities, ManiPT achieves the best overall performance.

## C.3. Hyperparameter Sensitivity

## C.4. Sensitivity to Consistency Constraint

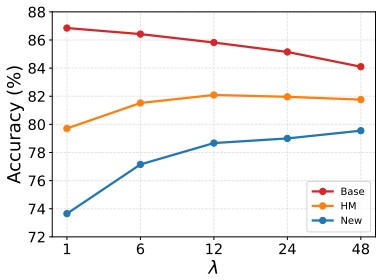

*Figure 5.* Sensitivity analysis of the consistency constraint weight.

We vary the weight $\lambda$ in Eq. (14). Figure 5 shows a clear trade off. When $\lambda$ is too small, the consistency constraint is insufficient to constrain feature adaptation, leading to manifold drift. When $\lambda$ is too large, the adaptation is over-confined to initial geometry, restricting the model's ability to adjust along transferable directions for task adaptation.

## C.5. Sensitivity to Context Length

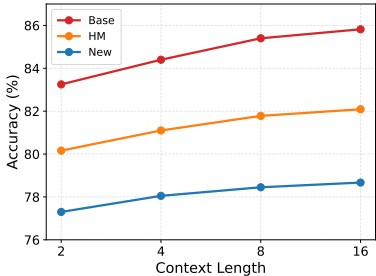

*Figure 6.* Sensitivity analysis of the context length. Unlike standard prompt tuning where increasing parameters often leads to overfitting, ManiPT effectively utilizes the increased capacity to boost both Base and Novel performance.

Figure 6 illustrates the impact of context length. We observe a consistent improvement in both Base and Novel accuracy as $m$ increases from 2 to 16. Typically, a larger context length increases the risk of overfitting to base classes, thereby hurting novel class generalization. However, ManiPT breaks this trend. This is due to the synergy between the manifold consistency constraints and the structural bias. The consistency constraints confine the feature adaptation within the pretrained geometric neighborhood, while the structural bias, via a geometric contraction mechanism, enforces incremental corrections. This dual safeguard enables the model to safely exploit the increased capacity to capture richer domain-specific semantics without overfitting to shortcut subspaces as capacity grows.

## C.6. Training and Inference sEfficiency

We evaluate the computational efficiency on an NVIDIA RTX 5090 GPU. The training time per epoch and peak GPU memory are measured on the ImageNet dataset under the base-to-novel setting, following the default configurations of each codebase. Note that for static prompt methods, the text features are pre-computed and cached offline. For inference latency, we report the per-image processing time averaged over 100 forward passes using a random input tensor of shape $1 \times 3 \times 224 \times 224$, following a warm-up phase of 50 iterations to ensure stable measurements. As shown in Table 16, ManiPT achieves a superior balance between efficiency and performance. In terms of training, it is exceptionally fast, requiring only 34.23 seconds per epoch. This is comparable to the lightweight baseline CoOp and approximately $7\times$ faster than MaPLe and $24\times$ faster than CoCoOp. Furthermore, ManiPT demonstrates remarkable parameter efficiency by training only the context vectors inserted at each layer. It utilizes just 0.25 M parameters which is an order of magnitude fewer than other deep prompting methods like MaPLe with 3.56 M parameters. Regarding inference, ManiPT maintains a low latency of 6.96 ms/img. While slightly higher than single-branch methods due to the dual-branch fusion, it remains highly suitable

*Table 16.* Training and inference efficiency comparison. OOM denotes out of memory.

| Method | Training Time (s/epoch) | Peak Memory (GB) | Infer. Latency (ms/img) | Params (M) |
|---|---|---|---|---|
| Zero-shot CLIP | N/A | N/A | 3.42 | 0.00 |
| CoOp | 29.12 | 8.33 | 3.50 | 0.01 |
| CoCoOp | 836.08 | 8.25 | 39.97 | 0.04 |
| MaPLe | 241.51 | 8.30 | 3.83 | 3.56 |
| PromptSRC | 248.49 | 8.63 | 3.87 | 0.05 |
| CoPrompt | 255.22 | 8.84 | 4.07 | 5.01 |
| TAC | 131.33 | 10.46 | 4.20 | 0.73 |
| TAP | N/A | OOM | 3041.71 | 3.73 |
| LLaMP | N/A | OOM | 4.19 | 52.77 |
| ManiPT | 34.23 | 10.23 | 6.96 | 0.25 |

for real-time deployment and is significantly more efficient than instance-conditional methods like CoCoOp. It is also worth noting that TAP and LLaMP require substantial GPU memory, often exceeding the capacity of commodity hardware. In particular, TAP's dynamic caption retrieval mechanism leads to extremely high inference latency, making it ill-suited for real-time deployment.

# D. Appendix 4: Manifold Drift Metric

## D.1. Definition

In this section, we provide the detailed computation procedure for the manifold drift metric. The metric quantifies the degree to which the features learned by prompt tuning deviate from the intrinsic geometric structure established by the pretrained CLIP model.

Let $\{\mathbf{x}_i\}_{i=1}^N$ denote the set of evaluation samples. Here, $N$ denotes the number of evaluation samples. We extract two sets of normalized feature representations. Pretrained features $\mathbf{Z}$ are generated by the frozen CLIP encoder. Let $\mathbf{Z} \in \mathbb{R}^{N \times d}$ denote the matrix whose $i$-th row is $\left(\mathbf{z}_{\mathbf{x}_i}^{\mathrm{vis}}\right)^\top$. Prompt-tuned features $\mathbf{H}$ are generated by the final fused representations. Let $\mathbf{H} \in \mathbb{R}^{N \times d}$ denote the matrix whose $i$-th row is $\left(\mathbf{f}_{\mathbf{x}_i}^{\mathrm{vis}}\right)^\top$.

Since the true nonlinear manifold of the pretrained model is analytically inaccessible, we approximate its local tangent space using PCA on the pretrained feature point cloud $\mathbf{Z}$.

We first compute the centroid of the pretrained features

$$\boldsymbol{\mu} = \frac{1}{N} \sum_{i=1}^N \mathbf{z}_{\mathbf{x}_i}^{\mathrm{vis}}. \tag{15}$$

Based on this centroid, we construct the centered data matrix with $\mathbf{1} \in \mathbb{R}^N$ denoting the all-ones vector.

$$\bar{\mathbf{Z}} = \mathbf{Z} - \mathbf{1}\boldsymbol{\mu}^\top. \tag{16}$$

We then perform Singular Value Decomposition (SVD) on $\bar{\mathbf{Z}}$ to obtain the principal directions

$$\bar{\mathbf{Z}} = \mathbf{U}\boldsymbol{\Sigma}\mathbf{V}^\top, \tag{17}$$

where $\mathbf{U} \in \mathbb{R}^{N \times d}$ contains the left singular vectors, $\boldsymbol{\Sigma} \in \mathbb{R}^{d \times d}$ is diagonal with singular values, and $\mathbf{V} \in \mathbb{R}^{d \times d}$ contains the right singular vectors sorted by singular value magnitude. We define the pretrained principal subspace $\mathcal{M}^{\mathrm{pca}}$ as the subspace spanned by the $d^{\mathrm{pca}}$ leading principal components. The projection matrix onto this subspace is given by

$$\mathbf{P}^{\mathrm{pca}} = \mathbf{V}^{\mathrm{pca}} \left(\mathbf{V}^{\mathrm{pca}}\right)^\top, \tag{18}$$

where $\mathbf{V}^{\mathrm{pca}} \in \mathbb{R}^{d \times d^{\mathrm{pca}}}$ denotes the first $d^{\mathrm{pca}}$ columns of $\mathbf{V}$. In our experiments, we set $d^{\mathrm{pca}} = 64$ to capture the dominant semantic directions.

Manifold drift is defined as the increase in the proportion of energy orthogonal to the principal subspace in the prompt-tuned features relative to the pretrained features. We define the ratio function $\mathcal{R}(\mathbf{Z})$ for a feature set $\mathbf{Z}$ relative to the pretrained center $\boldsymbol{\mu}$ and subspace $\mathbf{P}^{\mathrm{pca}}$

$$\mathcal{R}(\mathbf{Z}) = \frac{\sum_{i=1}^{N} \left\| [\mathbf{I}_d - \mathbf{P}^{\mathrm{pca}}] [\mathbf{z}_i - \boldsymbol{\mu}] \right\|_2^2}{\sum_{i=1}^{N} \| \mathbf{z}_i - \boldsymbol{\mu} \|_2^2 + \varepsilon^{\mathrm{num}}}, \tag{19}$$

where $\mathbf{z}_i$ is the $i$-th row of $\mathbf{Z}$, and $\varepsilon^{\mathrm{num}} = 10^{-12}$ is a numerical stability constant. The numerator represents the variance orthogonal to the pretrained manifold, and the denominator represents the total variance.

With these definitions in place, we compute the manifold drift $\Delta$ as

$$\Delta = \mathcal{R}(\mathbf{H}) - \mathcal{R}(\mathbf{Z}). \tag{20}$$

A positive value $\Delta > 0$ indicates that prompt tuning has shifted the feature distribution into directions that were previously low variance in the pretrained model, signifying a deviation from the pretrained manifold. The computation procedure is summarized in Algorithm 2.

---

**Algorithm 2** Computation of Manifold Drift $\Delta$

---

1: **Input:** Pretrained features $\mathbf{Z}$, Prompt-tuned features $\mathbf{H}$, PCA rank $d^{\mathrm{pca}}$.
2: **Output:** Manifold drift $\Delta$.
3: # 1. Approximating the Pretrained Manifold
4: Compute centroid $\boldsymbol{\mu}$ using Eq. (15)
5: Construct the centered data matrix $\bar{\mathbf{Z}}$ using Eq. (16)
6: Perform SVD on $\bar{\mathbf{Z}}$ to obtain the principal directions using Eq. (17)
7: Compute the projection matrix $\mathbf{P}^{\mathrm{pca}}$ using Eq. (18)
8: # 2. Computing the Ratio Function $\mathcal{R}(\mathbf{Z})$
9: Construct the centered data matrix $\bar{\mathbf{H}} \leftarrow \mathbf{H} - \mathbf{1}\boldsymbol{\mu}^{\top}$
10: Compute $\mathcal{R}(\mathbf{Z})$ using Eq. (19)
11: Compute $\mathcal{R}(\mathbf{H})$ using Eq. (19)
12: # 3. Computing Drift
13: Compute $\Delta$ using Eq. (20)

---

## D.2. Sensitivity to PCA Rank $d^{\mathrm{pca}}$

*Table 17.* The manifold drift metric $\Delta$ with respect to the PCA rank $d^{\mathrm{pca}}$ across all 11 datasets.

| Dataset | MaPLe | | | | ManiPT | | | |
|---|---|---|---|---|---|---|---|---|
| | $d^{\mathrm{pca}} = 32$ | $d^{\mathrm{pca}} = 64$ | $d^{\mathrm{pca}} = 128$ | $d^{\mathrm{pca}} = 256$ | $d^{\mathrm{pca}} = 32$ | $d^{\mathrm{pca}} = 64$ | $d^{\mathrm{pca}} = 128$ | $d^{\mathrm{pca}} = 256$ |
| ImageNet | 0.1320 | 0.1250 | 0.1085 | 0.0750 | 0.0038 | 0.0045 | 0.0051 | 0.0055 |
| Caltech101 | 0.0980 | 0.0910 | 0.0750 | 0.0480 | -0.0025 | -0.0021 | -0.0010 | 0.0005 |
| OxfordPets | 0.0890 | 0.0840 | 0.0680 | 0.0420 | 0.0008 | 0.0012 | 0.0015 | 0.0020 |
| StanfordCars | 0.1585 | 0.1520 | 0.1350 | 0.0950 | 0.0072 | 0.0078 | 0.0082 | 0.0085 |
| Flowers102 | 0.1450 | 0.1380 | 0.1120 | 0.0750 | -0.0055 | -0.0048 | -0.0030 | -0.0010 |
| Food101 | 0.0780 | 0.0720 | 0.0620 | 0.0450 | 0.0018 | 0.0023 | 0.0028 | 0.0032 |
| SUN397 | 0.1350 | 0.1290 | 0.1100 | 0.0820 | 0.0035 | 0.0039 | 0.0042 | 0.0048 |
| UCF101 | 0.1780 | 0.1710 | 0.1450 | 0.1020 | 0.0048 | 0.0055 | 0.0065 | 0.0075 |
| DTD | 0.1130 | 0.1139 | 0.1063 | 0.0669 | -0.0025 | -0.0019 | 0.0009 | 0.0024 |
| EuroSAT | 0.1698 | 0.1448 | 0.0961 | 0.0454 | 0.0095 | 0.0103 | 0.0084 | 0.0050 |
| FGVCAircraft | 0.1452 | 0.1619 | 0.1627 | 0.1076 | -0.0102 | -0.0037 | 0.0042 | 0.0048 |
| Average | 0.1310 | 0.1257 | 0.1073 | 0.0713 | 0.0010 | 0.0020 | 0.0035 | 0.0039 |

Table 17 presents the results of the manifold drift metric $\Delta$ with respect to different PCA ranks $d^{\mathrm{pca}} \in \{32, 64, 128, 256\}$. The rank $d^{\mathrm{pca}}$ determines the dimensionality of the reference subspace used to approximate the pretrained manifold. A smaller $d^{\mathrm{pca}}$ defines a stricter manifold constraint, capturing only the most dominant semantic directions, whereas a larger $d^{\mathrm{pca}}$ defines a more permissive subspace, encompassing subtle variations and potential noise. We observe three key phenomena that corroborate our theoretical analysis. First, metric sensitivity: for MaPLe, the drift metric generally decreases as $d^{\mathrm{pca}}$ increases. This behavior is physically consistent with our approximation calculation of manifold drift. As

the dimensionality of the principal subspace increases, it covers a larger feature space, so even significantly drifted features are more likely to fall close to this expanded subspace, resulting in lower manifold drift metrics. Second, robustness of ManiPT: ManiPT exhibits remarkable stability across all ranks, with $\Delta$ consistently remaining near zero. This insensitivity indicates that our method constrains downstream task features to align with the intrinsic geometric structure of the frozen CLIP features. Whether measured against a strict $d^{\mathrm{pca}} = 32$ or a permissive $d^{\mathrm{pca}} = 256$ manifold approximation, ManiPT maintains high consistency, confirming that the learned prompts do not undergo orthogonal deviation during semantic adjustment. Third, domain-specific trends: the behavior of MaPLe reveals that overfitting manifests differently across domains. On general datasets such as Caltech101, the drift drops rapidly at $d^{\mathrm{pca}} = 256$, suggesting that the deviation lies primarily in high-frequency variance directions. However, on fine-grained tasks like FGVCAircraft, the drift remains substantial even at $d^{\mathrm{pca}} = 256$. This implies that without our imposed consistency constraints and structural bias, prompt tuning in fine-grained low-data regimes tends to drive features toward directions structurally orthogonal to the pretrained manifold structure, and these directions likely correspond to shortcut features that cannot be captured by the low-dimensional approximation of the pretrained distribution.

Based on these empirical observations, we adopt $d^{\mathrm{pca}} = 64$ as the default value for our main experiments. It captures the dominant semantic components of the CLIP representation while being compact enough to sensitively detect semantic drift, whereas $d^{\mathrm{pca}} = 32$ may be too restrictive, leading to manifold underfitting, and $d^{\mathrm{pca}} = 256$ is too permissive, reducing the metric's discriminative power against noise-induced drift.

## E. Appendix 5: Limitations

Experimental results show that ManiPT generalizes across multiple settings. However, the framework has limitations inherent to its design. First, the structural bias, which enforces incremental corrections through dual-branch fusion, prioritizes geometric stability over raw inference speed, leading to increased latency compared to single-branch methods. Second, since LLM-generated descriptions serve as the geometric center for the textual consistency constraint, the effectiveness of this semantic anchoring depends on the quality of the external knowledge source, particularly in specialized narrow domains. Finally, the core premise of ManiPT is to confine the learned representations within the pretrained geometric neighborhood. While this prevents drift, it assumes that the pretrained manifold provides valid support for the downstream task. In scenarios with extreme distribution shifts where the optimal solution lies far outside the pretrained neighborhood, this geometric confinement might overly restrict the feature adaptation, necessitating a careful trade-off between manifold preservation and task-specific plasticity.

## F. Appendix 6: Notation

We summarize the main symbols used in the paper for clarity and consistency.

*Table 18.* Notation for core sets, indices, and scalars.

| Symbol | Description |
|---|---|
| $\mathcal{X}, \mathcal{Y}$ | Input image space and label space |
| $\mathcal{C} = \{1, \ldots, C\}$ | Class index set, $C$ denotes the number of classes |
| $\mathcal{D}$ | Data distribution over $\mathcal{X} \times \mathcal{Y}$ |
| $\mathcal{D}^{\mathrm{train}} = \{(\mathbf{x}_i, y_i)\}_{i=1}^N$ | Training dataset, $N$ denotes the number of samples |
| $T^{\mathrm{all}}, T_c^{\star}$ | Sets of all and class-specific token sequences for the text encoder |
| $d$ | Embedding dimension of CLIP features |
| $m$ | Number of learnable context vectors (prompt length) |
| $n$ | Number of LLM-generated descriptions per class |
| $K$ | Number of Transformer layers in the CLIP encoders |
| $d^{\mathrm{pca}}$ | PCA rank used to approximate the pretrained manifold |
| $i, j, c, k$ | Indices for samples, descriptions, classes, and layers, respectively |
| $\tau$ | Temperature parameter in the softmax logits |
| $\lambda$ | Weight of the consistency regularization |
| $E$ | Number of training epochs |
| $\rho$ | Confidence level in the generalization bounds |
| $B$ | Uniform upper bound on the cross-entropy loss |
| $\Lambda^{\mathcal{H}}$ | Lipschitz constant in the localized complexity analysis |
| $R^{\boldsymbol{\theta}}$ | Radius of the base parameter set for $\boldsymbol{\theta}$ |
| $D^{\boldsymbol{\theta}}$ | Dimension of the prompt parameter vector in the generalization analysis |
| $\varepsilon^{\mathrm{num}}$ | Numerical stability constant in the drift ratio |
| $\kappa^{\mathrm{vis}}, \kappa^{\mathrm{txt}}$ | Non-near-opposition margins for visual and text fusion |
| $\zeta^{\mathrm{max}}, \varepsilon^{\mathrm{proj}}$ | Maximum anchor neighborhood radius and average projection error bound |
| $\hat{d}_c$ | Distance between anchor $\mathbf{w}_c$ and its discrete projection $\tilde{\mathbf{w}}_c$ |

*Table 19.* Notation for features, prompts, and manifolds.

| Symbol | Description |
|---|---|
| $\mathcal{T}$, $\mathcal{V}$ | Frozen CLIP text and image encoders (prompts are provided as additional inputs during tuning) |
| $\mathbf{z}_c^{\text{txt}}$ | Frozen CLIP text feature for class $c$ |
| $\mathbf{z}_{\mathbf{x}}^{\text{vis}}$ | Frozen CLIP visual feature for image $\mathbf{x}$ |
| $\mathbf{h}_c^{\text{txt}}$ | Prompt-branch text feature for class $c$ |
| $\mathbf{h}_{\mathbf{x}}^{\text{vis}}$ | Prompt-branch visual feature for image $\mathbf{x}$ |
| $\mathbf{f}_c^{\text{txt}}$ | Final fused text feature for class $c$ |
| $\mathbf{f}_{\mathbf{x}}^{\text{vis}}$ | Final fused visual feature for image $\mathbf{x}$ |
| $\mathcal{P}^{\text{txt}}$, $\mathcal{P}^{\text{vis}}$ | Sets of deep text and visual prompt parameters |
| $A_c$, $A$ | LLM-generated description sets per class and over all classes |
| $E_c$, $\mathbf{w}_c$ | LLM-based description feature set and semantic prototype for class $c$ |
| $\bar{\mathbf{w}}_c$ | Discrete projection of $\mathbf{w}_c$ onto $\mathcal{M}_c^{\star,\text{txt}}$ |
| $\mathbf{F}^{\text{txt}}$, $\mathbf{W}$ | Matrices whose columns are class text features and semantic prototypes, respectively |
| $\mathbf{Z}$, $\mathbf{H}$ | Pretrained and prompt-tuned feature matrices in $\mathbb{R}^{N \times d}$ |
| $\bar{\mathbf{Z}}$, $\bar{\mathbf{H}}$ | Centered pretrained and prompt-tuned feature matrices |
| $\mathbf{U}$, $\boldsymbol{\Sigma}$, $\mathbf{V}$ | Matrix factors in the SVD of $\bar{\mathbf{Z}}$ |
| $\boldsymbol{\mu}$ | Mean of pretrained features used for centering |
| $\mathbf{V}^{\text{pca}}$ | Top $d^{\text{pca}}$ principal directions of pretrained features |
| $\mathbf{P}^{\text{pca}}$ | Projection matrix onto the PCA subspace |
| $\mathcal{Q}$, $\mathcal{S}$ | Transferable and shortcut feature subspaces |
| $\boldsymbol{\delta}$, $\boldsymbol{\delta}^{\mathcal{Q}}$, $\boldsymbol{\delta}^{\mathcal{S}}$ | Prompt-induced feature shift and its decomposition along $\mathcal{Q}$ and $\mathcal{S}$ |
| $\mathcal{M}^{\text{vis}}$ | Manifold of frozen CLIP visual features |
| $\mathcal{M}^{\text{txt}}$, $\mathcal{M}_c^{\star,\text{txt}}$ | Global and class-specific realizable text feature sets |

*Table 20.* Notation for losses and drift-related quantities.

| Symbol | Description |
|---|---|
| $\ell_c(\mathbf{x})$ | Logit for class $c$ given input $\mathbf{x}$ |
| $p(c \mid \mathbf{x})$ | Predicted class probability for input $\mathbf{x}$ |
| $\mathcal{L}^{\text{ce}}$ | Cross-entropy classification loss |
| $\mathcal{L}^{\text{img}}$ | Visual-side cosine consistency loss |
| $\mathcal{L}^{\text{txt}}$ | Text-side cosine consistency loss |
| $\mathcal{L}^{\text{con}}$ | Total consistency loss $\mathcal{L}^{\text{img}} + \mathcal{L}^{\text{txt}}$ |
| $\mathcal{L}^{\text{total}}$ | Overall ManiPT training objective function |
| $R(\boldsymbol{\theta})$ | Population risk with respect to distribution $\mathcal{D}$ |
| $\widehat{R}_N(\boldsymbol{\theta})$ | Empirical risk over $N$ training samples |
| $\mathcal{R}(\mathbf{Z})$ | Fraction of off-manifold energy for feature set $\mathbf{Z}$ |
| $\Delta$ | Manifold drift between prompt-tuned and pretrained features |
| $\Delta\boldsymbol{\ell}_{\mathbf{x}}^{\boldsymbol{\theta}}$ | Logit perturbation vector relative to the frozen model |
| $J^{\text{emp}}(h)$ | Empirical $\ell_2$ pseudo-metric on functions evaluated on the training sample |

