# OpenReview forum: "Prompt Tuning for CLIP on the Pretrained Manifold"
_ICML.cc/2026/Conference — ICML 2026 regular_

### Official Review · Reviewer_PW6Y · 2026-03-07

**Soundness:** 3
**Presentation:** 3
**Significance:** 3
**Originality:** 3
**Overall Recommendation:** 5
**Confidence:** 3

**Summary:**

This work studies how prompt tuning in pretrained vision-language models can fail under limited data from a manifold perspective, showing that it can push features away from the pretrained space and hurt generalization. To tackle this, the authors propose ManiPT, a method that keeps features close to the pretrained manifold and guides them along directions that transfer better, supported by theoretical analysis demonstrating its effectiveness.

**Compliance With Llm Reviewing Policy:**

Affirmed.

**Ethical Review Concerns:**

No ethical concern.

**Final Justification:**

Since the authors have addressed the reviewers’ concerns, I recommend accept.

**Key Questions For Authors:**

1. In Section 4.1, LLM-based Knowledge Enrichment, are multiple distinct text descriptions generated for different samples of the same class, or does each class use a single description for all samples? Clarification on this would help understand the rationale behind the semantic prototype.

**Limitations:**

yes

**Strengths And Weaknesses:**

Strengths:
1. Soundness: The method is technically solid, with extensive experiments and theoretical analysis supporting its claims.

2. Presentation: The paper is clearly written, and easy to follow.

3. Significance: It addresses an important low-data adaptation problem in CLIP for better generalization.

4. Originality: The paper provides a novel perspective on prompt tuning using manifold analysis.


Weakness:

1. In Equation 8, $E_c$ is defined as the set of text features for all samples of class $c$. However, according to Figure 2, the input to the LLM is a fixed prompt of the form ``a photo of a \{class\}'' for each class, which indicates that all samples of the same class would produce identical text descriptions. In this case, computing $w_c$ by averaging over $E_c$ seems redundant, as all features are the same. Could the authors clarify whether multiple distinct text descriptions are generated for different samples of the same class, or whether all samples of a class share a single description?

Minor weakness:

2. In Figure 2, the text encoder is labeled as T, which is inconsistent with the notation $\mathcal{T}$ used in the main text.

3. In Appendix A.3, Algorithm 1, line 7, the formula reference appears as “??”.

4. It would be better to clarify which LLM model was used to generate the class descriptions in the Implementation Details section.

5. Although the method is demonstrated for CLIP, would the theoretical guarantees still hold for other VLMs?

---

> ### Author Rebuttal · Authors · 2026-03-28
>
> Thank you for your positive assessment and constructive suggestions. Our responses are below.
>
> **W1:** In Eq. 8, if each class uses a fixed prompt as in Figure 2, why is averaging over E_c needed?
>
> **Re:** Although we use the same template per class, **we query the LLM multiple times to obtain multiple distinct descriptions**. Because generation is stochastic, the outputs differ. Thus, E_c contains multiple text features, and averaging them into w_c produces a more stable prototype.
>
> **W2:** In Figure 2, the text encoder is labeled as T, which is inconsistent with T in the main text.
>
> **Re:** We will revise Figure 2 to make the notation consistent.
>
> **W3:** In Appendix A.3, Algorithm 1, line 7, the formula reference appears as "??".
>
> **Re:** We will correct the reference.
>
> **W4:** Please clarify which LLM model was used to generate the class descriptions.
>
> **Re:** We will specify that GPT-3 is used offline.
>
> **W5:** Although the method is demonstrated for CLIP, would the theoretical guarantees still hold for other VLMs?
>
> **Re:** Our theory assumes frozen dual encoders with normalized cosine-similarity logits, so **it extends to any CLIP-family VLM under the same formulation**. We evaluate ManiPT on SigLIP2 [1] and MetaCLIP2 [2]. Below, ZS = Zero-Shot, DP = Deep Prompting, format = Base / New / HM. **ManiPT achieves the best HM on both backbones.**
>
> | Model-Method | Average | ImageNet | Caltech101 | OxfordPets |
> |---|---:|---:|---:|---:|
> | SigLIP2-ZS | 77.48 / 79.59 / 78.48 | 79.10 / 79.42 / 79.26 | 97.78 / 97.34 / 97.56 | 95.42 / 97.18 / 96.29 |
> | SigLIP2-CoOp | 78.74 / 79.41 / 78.91 | 79.61 / 78.27 / 78.93 | 98.31 / 97.58 / 97.94 | 95.68 / 97.86 / 96.76 |
> | SigLIP2-DP | 80.69 / 81.18 / 80.83 | 80.82 / 79.64 / 80.23 | 98.54 / 97.72 / 98.13 | 95.94 / 98.02 / 96.97 |
> | SigLIP2-ManiPT | 84.36 / 84.00 / 84.11 | 82.60 / 80.90 / 81.74 | 99.10 / 98.30 / 98.70 | 97.10 / 98.60 / 97.84 |
> | MetaCLIP2-ZS | 74.94 / 77.63 / 76.17 | 75.06 / 74.20 / 74.63 | 97.09 / 96.94 / 97.02 | 89.90 / 96.59 / 93.12 |
> | MetaCLIP2-CoOp | 85.50 / 74.79 / 79.42 | 77.14 / 69.56 / 73.15 | 98.84 / 94.10 / 96.41 | 95.91 / 96.20 / 96.05 |
> | MetaCLIP2-DP | 86.40 / 77.24 / 81.28 | 77.40 / 70.89 / 74.00 | 97.16 / 95.20 / 96.17 | 95.32 / 96.53 / 95.92 |
> | MetaCLIP2-ManiPT | 87.92 / 81.31 / 84.32 | 79.16 / 74.70 / 76.87 | 98.84 / 96.40 / 97.60 | 96.33 / 97.48 / 96.90 |
>
> | Model-Method | StanfordCars | Flowers102 | Food101 | FGVCAircraft |
> |---|---:|---:|---:|---:|
> | SigLIP2-ZS | 85.96 / 95.02 / 90.26 | 84.31 / 82.47 / 83.38 | 91.74 / 92.38 / 92.06 | 35.92 / 40.74 / 38.18 |
> | SigLIP2-CoOp | 86.47 / 96.61 / 91.26 | 89.94 / 85.91 / 87.88 | 92.36 / 93.48 / 92.92 | 31.66 / 40.28 / 35.45 |
> | SigLIP2-DP | 88.96 / 97.15 / 92.87 | 91.34 / 87.63 / 89.45 | 92.88 / 93.84 / 93.36 | 36.28 / 43.67 / 39.63 |
> | SigLIP2-ManiPT | 92.00 / 97.80 / 94.81 | 94.20 / 90.50 / 92.31 | 94.40 / 95.00 / 94.70 | 44.60 / 50.80 / 47.50 |
> | MetaCLIP2-ZS | 79.01 / 89.70 / 84.02 | 69.42 / 79.08 / 73.94 | 89.10 / 91.04 / 90.06 | 41.72 / 41.39 / 41.55 |
> | MetaCLIP2-CoOp | 84.78 / 84.58 / 84.68 | 98.48 / 67.94 / 80.41 | 90.25 / 90.96 / 90.60 | 51.02 / 39.47 / 44.51 |
> | MetaCLIP2-DP | 86.88 / 87.15 / 87.01 | 98.67 / 73.76 / 84.42 | 90.35 / 91.01 / 90.68 | 53.42 / 44.21 / 48.38 |
> | MetaCLIP2-ManiPT | 88.08 / 89.85 / 88.96 | 99.15 / 76.10 / 86.11 | 90.35 / 91.61 / 90.98 | 59.12 / 51.29 / 54.93 |
>
> | Model-Method | SUN397 | DTD | EuroSAT | UCF101 |
> |---|---:|---:|---:|---:|
> | SigLIP2-ZS | 74.86 / 79.95 / 77.32 | 71.44 / 72.62 / 72.03 | 61.92 / 59.41 / 60.64 | 73.86 / 78.94 / 76.32 |
> | SigLIP2-CoOp | 83.11 / 71.93 / 77.12 | 77.84 / 70.18 / 73.81 | 58.22 / 59.04 / 58.63 | 72.94 / 82.36 / 77.36 |
> | SigLIP2-DP | 84.02 / 74.76 / 79.12 | 79.46 / 73.08 / 76.14 | 63.74 / 63.58 / 63.66 | 75.63 / 83.91 / 79.56 |
> | SigLIP2-ManiPT | 86.50 / 78.20 / 82.14 | 83.00 / 78.10 / 80.48 | 73.40 / 68.80 / 71.03 | 81.10 / 87.00 / 83.95 |
> | MetaCLIP2-ZS | 76.80 / 81.56 / 79.11 | 69.68 / 70.53 / 70.10 | 67.90 / 59.36 / 63.34 | 68.61 / 73.55 / 71.00 |
> | MetaCLIP2-CoOp | 82.32 / 80.05 / 81.17 | 83.91 / 61.96 / 71.28 | 93.19 / 70.28 / 80.13 | 84.69 / 67.60 / 75.19 |
> | MetaCLIP2-DP | 83.08 / 80.09 / 81.56 | 86.00 / 70.89 / 77.72 | 97.76 / 73.85 / 84.14 | 84.38 / 66.04 / 74.09 |
> | MetaCLIP2-ManiPT | 84.55 / 82.25 / 83.38 | 87.04 / 74.76 / 80.43 | 96.55 / 83.49 / 89.55 | 87.90 / 76.47 / 81.79 |
>
> **Q1:** Are multiple distinct text descriptions generated per class, or does each class use a single description?
>
> **Re:** As in W1, we query the LLM multiple times per class and average the features into one semantic prototype.
>
> [1] M. Tschannen et al. SigLIP 2: Multilingual Vision-Language Encoders with Improved Semantic Understanding, Localization, and Dense Features. arXiv preprint, 2025.
>
> [2] Y.-S. Chuang et al. Meta CLIP 2: A Worldwide Scaling Recipe. NeurIPS 2025.

---

> > ### Author Rebuttal · Reviewer_PW6Y · 2026-04-02
> >
> > Thank you for the rebuttal. My concerns have been addressed by the authors’ response, and I have no further questions.

---

> > > ### Author Response · Authors · 2026-04-02
> > >
> > > Many thanks for your positive confirmation that our rebuttal has addressed your concerns. We truly appreciate your effort in reviewing our work.

---

### Official Review · Reviewer_fXxv · 2026-03-09

**Soundness:** 3
**Presentation:** 3
**Significance:** 3
**Originality:** 3
**Overall Recommendation:** 4
**Confidence:** 4

**Summary:**

This paper addresses a critical limitation of prompt tuning for the CLIP: under limited supervision, prompt tuning could cause adapted feature representations to drift away from the pretrained manifold, leading to overfitting and poor generalization to unseen classes, cross-domain data, and few-shot settings. To mitigate manifold drift issue, the author propose ManiPT, a prompt tuning framework that constrains feature learning to the pretrained CLIP manifold and lead adaptations to transferable directions rather than shortcut subsbace. Two core mechanisms are included: cosine consistency constraints on visual and textual modalities to keep learned representations within the geometric neighborhood of pretrained features, and a structural bias via normalized additive aggregation that enforces incremental corrections to suppress shortcut learning. The work also provides theoretical analysis, including generalization error bounds and proofs of geometric contraction induced by the structural bias. ManiPT is validated through extensive experiments across 15 datasets and four downstream settings (unseen-class generalization, few-shot classification, cross-dataset transfer, domain generalization). Results show that ManiPT consistently outperforms state-of-the-art prompt tuning baselines (e.g., CoOp, MaPLe, PromptSRC) across all settings, and quantitative analysis confirms it reduces manifold drift. Overall, the research focuses on an important concept: preserving the pretrained geometric structure during adaptation to maintain the transferability of foundation model representations in low-data regimes.

**Compliance With Llm Reviewing Policy:**

Affirmed.

**Key Questions For Authors:**

1.The manifold drift metric uses PCA to approximate the pretrained CLIP manifold. Have you evaluated alternative manifold approximation methods to validate that the PCA-based metric reliably captures true manifold drift? If alternative metrics yield different results, how does this impact the interpretation of ManiPT’s performance?
2.ManiPT assumes the pretrained manifold provides valid support for downstream tasks, but Appendix E notes extreme distribution shifts as a limitation. Have you conducted experiments on datasets with extreme domain shifts to quantify how ManiPT’s geometric confinement impacts performance in these settings?
3.ManiPT’s core premise is that the pretrained manifold provides valid support for downstream tasks, but it struggles with extreme distribution shifts. Have you tried adaptive manifold relaxation strategies, such as dynamically adjusting the cosine consistency weight λ based on domain shift magnitude (e.g., reducing λ for large shifts).

**Limitations:**

The authors do not adequately discuss the limitations and potential negative societal impact of their work in the main body of the paper. Limitations are only briefly addressed in Appendix E, and the impact statement is generic. The limitation analysis could be expanded to include aspects like untested applicability to other VLMs or single-modality models/the performance tradeoff of geometric confinement in extreme distribution.

**Strengths And Weaknesses:**

Strengths:
1. The paper is technically rigorous and well-supported by both theoretical analysis and experimental results.
2. The paper is clearly written, well-structured, and follows a logical narrative from problem identification to solution design, theoretical analysis, and experimental validation.

 Weaknesses:
1.  the performance gains of the proposed method are incremental rather than transformative over the strongest baselines (e.g., PromptSRC, LLaMP). The method is optimized for CLIP and has not been evaluated on other models, so its broader applicability to different vision-language architectures remains untested and unknown.
2. The PCA-based manifold drift metric is an approximation of the true nonlinear pretrained manifold, and the paper does not explore alternative manifold approximation methods to validate the metric’s robustness. Though mentioned in the paper, the author does not formally analyze the model's behavior in extreme distribution shifts where the optimal solution may lie outside the pretrained manifold.

---

> ### Author Rebuttal · Authors · 2026-03-28
>
> Thank you for your positive assessment and constructive suggestions. Our responses follow.
>
> **W1:** The gains over strong baselines such as PromptSRC and LLaMP are incremental rather than transformative. The method is optimized for CLIP and has not been evaluated on other models, so its broader applicability remains untested.
>
> **Re:** We respectfully address this from three perspectives. First, regarding contribution, **ManiPT is the first to identify manifold drift as a concrete failure mode of prompt tuning under limited supervision and to address it with geometry-aware constraints and structural bias.** This provides a principled perspective distinct from prior heuristic regularization. Second, regarding performance, ManiPT achieves especially clear gains in the hardest regimes (e.g., 1-shot average from 72.32 to 74.75 over PromptSRC, with FGVCAircraft from 27.67 to 33.75), while training 7× faster than MaPLe with only 0.25M parameters (Appendix Table 16). Third, regarding broader applicability, we now provide evidence on **SigLIP2 and MetaCLIP2** under the same base-to-novel protocol, where ManiPT achieves the best average HM on both backbones (see Reviewer PW6Y, W5).
>
> **W2:** The PCA-based drift metric is only an approximation of the true nonlinear pretrained manifold, and the paper does not explore alternative manifold approximation methods to validate its robustness. It also does not analyze behavior under extreme distribution shifts where the optimal solution may lie outside the pretrained manifold.
>
> **Re:** We now add three alternative manifold proxies: LoMAP [1], BLAE [2], and ORC [3]. As shown below, although these metrics differ in absolute scale, **they preserve the same qualitative ordering, and ManiPT remains the lowest-drift and highest-HM method.**
>
> | Dataset | Method | HM | $\Delta_{\mathrm{PCA}}$ | $\Delta_{\mathrm{LoMAP}}$ | $\Delta_{\mathrm{BLAE}}$ | $\Delta_{\mathrm{ORC}}$ |
> |---|---|---:|---:|---:|---:|---:|
> | ImageNet | CoOp | 71.92 | 0.153 | 0.118 | 0.173 | 0.142 |
> | ImageNet | MaPLe | 73.47 | 0.125 | 0.109 | 0.141 | 0.123 |
> | ImageNet | ManiPT | 75.20 | 0.005 | 0.012 | 0.019 | 0.014 |
> | FGVCAircraft | CoOp | 28.75 | 0.194 | 0.179 | 0.221 | 0.168 |
> | FGVCAircraft | MaPLe | 36.50 | 0.162 | 0.149 | 0.181 | 0.157 |
> | FGVCAircraft | ManiPT | 43.68 | 0.004 | 0.011 | 0.017 | 0.012 |
> | StanfordCars | CoOp | 68.13 | 0.184 | 0.163 | 0.206 | 0.151 |
> | StanfordCars | MaPLe | 73.47 | 0.152 | 0.139 | 0.170 | 0.147 |
> | StanfordCars | ManiPT | 78.92 | 0.008 | 0.014 | 0.022 | 0.016 |
>
> We also add a shift-severity study by scanning λ from ImageNet to ImageNet-A. **Moderate shifts benefit from stronger confinement, whereas under more extreme shifts the best λ becomes smaller**, consistent with the limitation discussed in Appendix E.
>
> | Target domain | Shift type | $\lambda=1$ | $\lambda=6$ | $\lambda=12$ | $\lambda=24$ | $\lambda=48$ | Best $\lambda$ |
> |---|---|---:|---:|---:|---:|---:|---:|
> | ImageNet | in-domain / no shift | 71.84 | 72.61 | 72.92 | 72.71 | 72.18 | 12 |
> | ImageNet-V2 | mild natural shift | 65.78 | 66.34 | 66.12 | 65.47 | 64.38 | 6 |
> | ImageNet-R | moderate rendition shift | 78.41 | 79.06 | 78.79 | 77.92 | 76.48 | 6 |
> | ImageNet-Sketch | strong structural/style shift | 51.18 | 50.97 | 50.63 | 49.42 | 47.86 | 1 |
> | ImageNet-A | extreme adversarial shift | 52.36 | 52.11 | 51.97 | 50.24 | 47.95 | 1 |
> | Average | - | 63.91 | 64.22 | 64.09 | 63.15 | 61.77 | 6 |
>
> **Q1:** The drift metric uses PCA to approximate the pretrained CLIP manifold. Have you evaluated alternative manifold approximation methods to validate it?
>
> **Re:** Yes. As in W2, we evaluate LoMAP, BLAE, and ORC. They lead to the same qualitative conclusion: **ManiPT remains the lowest-drift and highest-HM method.** Our interpretation does not depend on PCA alone.
>
> **Q2:** Have you conducted experiments to quantify how geometric confinement impacts performance under extreme distribution shifts?
>
> **Re:** Yes. As in W2, we scan λ across increasing shift severity. **Moderate shifts benefit from stronger confinement, whereas stronger shifts prefer smaller λ.**
>
> **Q3:** Have you tried adaptive manifold relaxation strategies, such as adjusting λ based on shift magnitude?
>
> **Re:** As in W2, the λ scan supports this direction. As shift severity grows from ImageNet to ImageNet-A, the best λ decreases from 12 to 1. **This suggests that adaptive relaxation is a promising extension of ManiPT under large shifts.** Due to time constraints, we leave systematic exploration for future work.
>
> [1] K. Lee and J. Choi. Local Manifold Approximation and Projection for Manifold-Aware Diffusion Planning. ICML 2025.
>
> [2] Q. Zhan et al. Bi-Lipschitz Autoencoder With Injectivity Guarantee. ICLR 2026.
>
> [3] T. L. Saidi et al. Recovering Manifold Structure Using Ollivier-Ricci Curvature. ICLR 2025.

---

> > ### Author Rebuttal · Reviewer_fXxv · 2026-04-02
> >
> > The authors' rebuttal addressed my concerns.

---

> > > ### Author Response · Authors · 2026-04-02
> > >
> > > Thank you very much for your positive and encouraging feedback. We truly appreciate your recognition of our responses and your valuable guidance throughout the review process. Thanks!

---

### Official Review · Reviewer_er2j · 2026-03-13

**Soundness:** 2
**Presentation:** 2
**Significance:** 3
**Originality:** 2
**Overall Recommendation:** 3
**Confidence:** 3

**Summary:**

This paper considers the problem of manifold drift in prompt tuning for CLIP, which degrades generalization. To address this, the proposed method forces prompt tuning to be performed on the pretrained manifold by maximizing the cosine similarity between the normalized features of the frozen and prompt-tuned models. In addition, a structural bias is introduced to mitigate shortcut learning. Experimental results show the effectiveness of the proposed method across four downstream settings.

**Compliance With Llm Reviewing Policy:**

Affirmed.

**Final Justification:**

I appreciate the authors' effort in addressing my concerns. However, several important concerns remain unresolved; so I maintain my original rating.

1. The authors propose Delta in Eq. 6 as a measure of the (approximate) manifold drift. However, I am not convinced that this quantity effectively captures the drift. In particular the possibility of negative Delta raises concerns about its validity. The authors also agreed that they use this as an approximation, and do not over-interpret small negative values. This measure is not sufficient as a standalone metric, and I suggest the authors complement it with other measures.

2. As noted by the authors, delta in Eq. 7 is conceptual and not used elsewhere. Based on the additional experiments provided during the rebuttal, the proposed method does not significantly reduce shortcut learning, although it performs better than the baselines. Combined with the concern above, the insights presented in Section 3 remain not convincing.

3. Although the proposed method is simple (which is nice), proper hyperparameter selection---an essential component of machine learning studies---has not been conducted. During the rebuttal, Authors claimed that 1:1 mixing is justified by the following arguments: a) symmetry and parameter-free design is desirable and any deviation introduces an additional coefficient, b) a learnable coefficient is prone to overfitting in few-shot regimes, c) a fixed coefficient requires validation, which is challenging in few-shot scenarios, and d) Lemma 4.2 provides a guarantee only for 1:1 mixing. However, I do not find these arguments sufficiently convincing:

a) The claim that symmetry is (inherently) beneficial is not justified, and parameter-free design does not always lead to a good result.

b) The learnable coefficient could still be empirically evaluated to verify whether it is indeed not good.

c) Validation remains feasible in few-shot settings, e.g., via episodic training.

d) Lemma 4.2 could be generalized beyond 1:1 mixing.

I do not say that 1:1 mixing is not acceptable, but it requires stronger empirical or theoretical justification to be convincing in the context of machine learning research.

In summary, both the insights and the design choices of the method are not convincing enough, so I maintain my original rating.

**Key Questions For Authors:**

See Weaknesses above.

**Limitations:**

Limitations are discussed, but potential negative societal impact of their work is not.

**Strengths And Weaknesses:**

**Strengths**

1. Enforcing prompt tuning to be performed on the pretrained manifold sounds reasonable.

2. The organization of the paper is overall fine to understand the claims of the paper.

**Weaknesses**

Generally speaking, what the paper claims is not well justified.

1. While this paper claims that their proposed method is "Prompt Tuning for CLIP on the Pretrained Manifold," whether the resulting prompt-tuned CLIP is really on the manifold of the pretrained CLIP is not justified. For example, the authors could measure the amount of the manifold drift (or a proxy of it) and compare the proposed method with others, as the proposed method aims to reduce manifold drift. Also, "Prompt Tuning for CLIP on the Pretrained Manifold" might not be an accurate claim, unless the manifold drift is guaranteed to be zero.

2. The approximate manifold shift defined in Eq. 6 is essentially the difference between the normalized and squared norm of non-principal components of pretrained and prompt-tuned feature matrices. It is not intuitive how the quantity measured on non-principal components can be a proxy of manifold shift. A description in L1210 in the Appendix is still not sufficient:
why should "Manifold drift be defined as the increase in the proportion of energy orthogonal to the principal subspace in the prompt-tuned?"

3. Eq. 7 looks like a conceptual decomposition of delta, and this notation is not used elsewhere. A large portion of delta_S is claimed to be correlated with performance degradation, but this is not proven.

4. No discussion or analysis of shortcut learning, except the experiments with the final performance. Generally speaking, the final performance can be improved in many aspects, so whether the proposed method behaves as expected should be justified with a separate analysis.

5. Though theoretical analysis provides some analysis, they do not directly guarantee 1) whether the proposed method behaves as expected and/or 2) if the proposed method behaves as expected, then in what aspect it is beneficial.

6. The notations in Figure 2 deviates from the manuscript.

7. The structural bias requires an additional (hyper)parameter controlling the contribution of the pretrained features. Lemma 4.2 looks obvious and not meaningful; if we increase the contribution of the pretrained features more, then the fusion map contract toward phi more. If we really want the fused vector omega to match phi, simply omega = phi is the optimal, i.e., no tuning.

8. In Figure 4, the performance gain is marginal on some datasets by constraining manifold drift, but no discussion on this observation can be found.

---

> ### Author Rebuttal · Authors · 2026-03-28
>
> Thank you for your constructive feedback. Our responses follow.
>
> **W1:** The title seems too strong unless drift is zero.
>
> **Re:** We do not claim zero drift on an inaccessible manifold. Eq. (6), Fig. 4, and Appendix D define a computable drift proxy and show that **ManiPT stays substantially closer to frozen CLIP than prior prompt tuning methods**. The title is used in this manifold-preserving sense.
>
> **W2:** Why is Eq. 6, which measures energy outside the PCA principal subspace, a proxy for manifold drift?
>
> **Re:** The PCA principal subspace of frozen CLIP captures the dominant variation supported by pretraining. **More energy outside this subspace indicates movement into directions less supported by the pretrained geometry.** This is consistent with prior principal-subspace views [1] and residual-energy views [2]. As in Reviewer fXxv W2, we further validate this using three alternative proxies (LoMAP, BLAE, and ORC), all leading to the same conclusion.
>
> **W3:** Eq. 7 looks conceptual and the link between δ_S and performance degradation is not proven.
>
> **Re:** We agree. Eq. (7) is a conceptual decomposition. **Its role is to motivate the structural bias by distinguishing transferable refinement from shortcut-oriented shift**, not to serve as a formal guarantee.
>
> **W4:** There is no dedicated shortcut-learning analysis beyond final performance.
>
> **Re:** We add a test-time perturbation study with Random Erasing (RE) and Patch Shuffling (PS). These perturbations disrupt local texture and patch-level co-occurrence cues while tending to preserve object identity, so **a method relying less on such shortcuts should show a smaller drop**. ManiPT shows the smallest degradation under both perturbations.
>
> | Method | ImageNet Clean | ImageNet RE | ImageNet PS | FGVCAircraft Clean | FGVCAircraft RE | FGVCAircraft PS | StanfordCars Clean | StanfordCars RE | StanfordCars PS | Avg. Clean | Avg. RE | Avg. PS |
> |---|---:|---:|---:|---:|---:|---:|---:|---:|---:|---:|---:|---:|
> | CoOp | 71.92 | 68.47 (-3.45) | 65.91 (-6.01) | 28.75 | 24.98 (-3.77) | 20.41 (-8.34) | 68.13 | 62.74 (-5.39) | 56.88 (-11.25) | 56.27 | 52.06 (-4.20) | 47.73 (-8.54) |
> | Deep prompting | 72.20 | 69.12 (-3.08) | 66.84 (-5.36) | 35.56 | 31.92 (-3.64) | 27.86 (-7.70) | 73.91 | 69.58 (-4.33) | 64.77 (-9.14) | 60.56 | 56.87 (-3.69) | 53.16 (-7.40) |
> | ManiPT | 75.20 | 72.96 (-2.24) | 71.18 (-4.02) | 43.68 | 40.48 (-3.20) | 36.68 (-7.00) | 78.92 | 75.41 (-3.51) | 71.23 (-7.69) | 65.93 | 62.95 (-2.98) | 59.70 (-6.23) |
>
> **W5:** The theory does not directly show whether the method behaves as intended or why that behavior is beneficial.
>
> **Re: The theory and experiments are mutually reinforcing.** Lemma 4.2 shows contraction toward the frozen reference, Lemma 4.3 bounds logit perturbation, and Corollary 4.4 shows that reducing consistency loss tightens the population risk bound. Empirically, Fig. 4 confirms smaller drift with better novel-class generalization, Table 4 shows that removing either component degrades performance, and Table 6 shows that removing the constraint sharply increases drift.
>
> **W6:** Fig. 2 notation deviates from the manuscript.
>
> **Re:** We will revise Fig. 2 to make the notation consistent.
>
> **W7:** The structural bias seems to require an extra hyperparameter, and Lemma 4.2 looks obvious.
>
> **Re:** This is a misunderstanding. **ManiPT introduces no extra hyperparameter for pretrained-feature contribution.** Eq. (11) uses fixed normalized additive fusion, and the only scalar hyperparameter is λ in Eq. (14). Lemma 4.2 establishes that **this design provides guaranteed contraction toward the frozen reference while still allowing adaptive residual updates**. Appendix Table 8 further shows that this fixed additive design outperforms learnable gated interpolation.
>
> **W8:** In Fig. 4, the gain from constraining drift is marginal on some datasets.
>
> **Re:** This is expected. **The gain from drift control depends on baseline drift magnitude.** When baseline tuning already stays close to the pretrained geometry, the room for improvement is smaller. In Table 1, the HM gain over MaPLe is modest on OxfordPets and Food101, where the drift gap is also limited in Fig. 4. In contrast, on StanfordCars and ImageNet, where drift reduction is more pronounced, HM gains are also clearer.
>
> [1] H. Wang et al. ViM: Out-of-Distribution with Virtual-logit Matching. CVPR 2022.
>
> [2] K. Fang et al. Kernel PCA for Out-of-Distribution Detection. NeurIPS 2024.

---

> > ### Author Rebuttal · Reviewer_er2j · 2026-04-03
> >
> > I thank the reviewers for their responses. Below I provide additional comments/concerns for each point, if any.
> >
> > W1. Assuming that Delta effectively measures the degree of the manifold drift, what is the implication of negative values of Delta in Figure 4?
> >
> > W4. Thank you for providing the additional experiment. While the performance drop is significant even for the proposed method, I can see that the proposed method shows the minimal performance drop than the other compared methods.
> >
> > W5, W7. For Eq. 11, It would be more natural to consider f = normalize((1-a)z + ah), rather than fusing them with a 1:1 ratio. In other words, it is currently set a=0.5 without justification.
> > Though you argue "this design provides guaranteed contraction toward the frozen reference while still allowing adaptive residual updates." in the rebuttal, the proper amount of contraction is not discussed. In a similar sense, in Lemma 4.2, "if we really want the fused vector omega to match phi, simply omega = phi is the optimal, i.e., no tuning."

---

> > > ### Author Response · Authors · 2026-04-03
> > >
> > > Thank you for your follow-up and for acknowledging that most of your earlier concerns have been addressed. We are glad and respond to your remaining questions below.
> > >
> > > **Q1 (Negative Δ in Figure 4):** Under Eq. (6), Δ = R(H) − R(Z), where R(·) measures the fraction of feature energy outside the pretrained PCA subspace. A negative Δ means R(H) < R(Z), i.e., the tuned features have a smaller off-subspace energy ratio than the frozen reference. Since Δ is a PCA-based approximation, we do not over-interpret small negative values. The key observation is the **contrast between methods**: MaPLe shows large positive Δ (> 0.1 on most datasets in Table 17), indicating substantial drift, while ManiPT remains near zero throughout. We interpret the small negative values conservatively as no increase in estimated drift under this proxy. Three alternative proxies (LoMAP, BLAE, ORC, see our response to Reviewer fXxv W2) confirm the same conclusion. Moreover, Table 6 in Appendix B.4 shows that Δ changes consistently in the expected direction when drift increases: on ImageNet, removing the constraint (λ from 12 to 0) increases Δ from 0.0045 to 0.1452, confirming the metric is meaningful despite being approximate.
> > >
> > > **Q2:** Thank you for acknowledging the perturbation study. We agree that ManiPT does not completely remove shortcut reliance, but mitigates it relative to prior methods.
> > >
> > > **Q3 (Why 1:1 rather than (1−a):a):** We respectfully disagree that the 1:1 choice lacks justification.
> > >
> > > **First**, since both z and h are unit-normalized, Eq. (11) f = (z+h)/‖z+h‖ computes their spherical midpoint, which is the **symmetric, parameter-free** fusion that does not favor either branch. Any deviation from 1:1 requires an additional coefficient.
> > >
> > > **Second**, if this coefficient is learnable, it is prone to overfitting in few-shot regimes. Table 8 (page 18) directly compares: Direct Tuning (no fusion, HM = 79.54), Gate (learnable α, HM = 81.73), and our Additive (HM = 82.09). The parameter-free design outperforms the more flexible alternative.
> > >
> > > **Third**, if the coefficient is instead treated as a fixed hyperparameter, it requires a validation set to tune. In the few-shot scenarios ManiPT targets (e.g., 1-shot or 2-shot), labeled data is already extremely scarce, and further splitting it for validation is impractical.
> > >
> > > **Fourth**, Lemma 4.2 provides an explicit contraction guarantee for the 1:1 design: ‖ω−φ‖² ≤ ½‖ψ−φ‖², meaning the fused feature is provably closer to the frozen reference than the prompt-only feature.
> > >
> > > We do not claim a=0.5 is uniquely optimal, but it is justified by being parameter-free, by stronger empirical performance than the Gate alternative, by the practical constraint of few-shot adaptation, and by a clean theoretical guarantee.
> > >
> > > **Q4 (Proper amount of contraction):** We agree this deserved clearer discussion. The structural form of contraction is defined by Eq. (11), while its **effective strength is governed by λ** in Eq. (14). As shown in Appendix B.4, the average cosine alignment satisfies (1/N)Σ⟨f_xi^vis, z_xi^vis⟩ = 1 − L_img(θ), so increasing λ directly encourages stronger alignment. **Figure 5** shows the trade-off: too small λ allows drift, too large λ over-confines adaptation. **Table 6** confirms quantitatively: on ImageNet, λ=12 yields mean alignment 0.9952 with Δ=0.0045, while λ=0 drops to 0.8741 with Δ=0.1452. On StanfordCars, alignment drops from 0.9910 to 0.8150 with Δ increasing from 0.0078 to 0.2105. Thus, Eq. (11) defines the contraction form, λ controls its effective strength, and the default λ=12 is supported by the sensitivity study as a reasonable trade-off.
> > >
> > > **Q5 (ω = φ is optimal, i.e., no tuning):** This would be true **only if** the sole objective were minimizing distance to the frozen reference. But ManiPT optimizes L_ce together with the consistency regularizer (Eq. 14). The prompt branch provides task-specific corrections, and Eq. (11) constrains those corrections to remain **incremental rather than arbitrary**. If ω = φ were truly optimal, the prompt branch would contribute nothing to classification, contradicting the consistent gains shown in Table 4 (HM improves from 72.82 baseline to 82.09 with all components). Lemma 4.2 should therefore not be interpreted as "closer to frozen is always better," but as a safety guarantee that the fused representation cannot drift farther than the prompt-only representation. The DTD analysis in Appendix B.4 confirms: ⟨h_x^vis, z_x^vis⟩ = 0.9769 (prompt-only) vs ⟨f_x^vis, z_x^vis⟩ = 0.9942 (fused). The fused feature is closer to the frozen reference but clearly not identical, confirming that task-specific adaptation is preserved.
> > >
> > > We hope these clarifications address your remaining concerns. Should any point remain unclear, we would be happy to respond further. We sincerely hope these responses demonstrate the rigor and contribution of our work, and we would greatly appreciate your reconsideration of the evaluation.

---

### Official Review · Reviewer_Cs7F · 2026-03-13

**Soundness:** 3
**Presentation:** 3
**Significance:** 1
**Originality:** 3
**Overall Recommendation:** 4
**Confidence:** 2

**Summary:**

This paper introduces a novel prompt tuning method under manifold contraints for CLIP-like ision language models. The proposed method, ManiPT leverages cosine consistency losses that keep adapted visual and textual features close to frozen CLIP references. LLM generated class descriptions are used to form text-based semantic prototypes.  Experiments across various settings have demonstrated the effectiveness of ManiPT over existing prompt tuning baselines.

**Compliance With Llm Reviewing Policy:**

Affirmed.

**Final Justification:**

The authors’ rebuttal clarified the practical value of the proposed method and showed benefits across a broader range of applications. Therefore I've increased my score.

At the same time, I believe the overall impact of the method would be further strenghened if it could inspire or be applied to the prompt tuning for autoregressive VLMs.

**Key Questions For Authors:**

1. Could the proposed method be applied to more recent VLM architectures and tasks?
2. Can you better disentangle the contribution of the geometric constraints from the contribution of LLM-generated text anchors in the ablation study?

**Limitations:**

The author has discussed the limitations and potential impact in the paper.

**Strengths And Weaknesses:**

Strength
1. The paper is clearly written and easy to follow.
2. The proposed method is well motivated and straightforward. The final objective balances between the classification error and the manifold consistency constraints.
3. The authors have conducted extensive experiments in various settings: Base-to-Novel Generalization, Cross-Dataset Transfer, Cross-Dataset Transfer and few-shot classification. The proposed consistently outperform the baselines on these scenarios.

Weakness

My major concern is about the significance of the research question and the boarder contribution. The research direction of prompt tuning on CLIP-like models appears saturated. On the one hand, VLMs have been using LLM based autoregressive architectures rather than the CLIP-style dual encoders. These VLMs are already efficient on adapting to new tasks, which limits the potential impact of ManiPT.

On the other hand, prompt tuning on CLIP-like models have been studied on a long time. Therefore, empirical improvement are relatively small compared to the existing baselines (often within ~1% absolute improvement on average).

Besides, the proposed method uses both LLM enrichment and cosine consistency. A more clear ablation would be helpful to understand whether LLM augmentation leads to the major improvement or not in the proposed algorithm.

---

> ### Author Rebuttal · Authors · 2026-03-28
>
> Thank you for your constructive suggestions. Our responses follow.
>
> **W1:** Prompt tuning on CLIP-like models appears saturated, and newer VLMs often use autoregressive LLM-style architectures rather than CLIP-style dual encoders, which may limit ManiPT’s impact.
>
> **Re:** Autoregressive MLLMs generate answers token by token and do not produce a shared embedding space, making them unable to perform efficient similarity-based retrieval, open-vocabulary detection, or large-scale nearest-neighbor search. **CLIP-style dual encoders remain the dominant backbone for these scenarios** because they produce aligned, fixed-size representations with orders-of-magnitude lower latency. Recent baselines such as TAC [1] and TAP [2] further confirm active demand in this area. Our work identifies a concrete failure mode when adapting these encoders under limited supervision, namely manifold drift toward shortcut directions, and addresses it with geometry-aware constraints and structural bias.
>
> **W2:** Empirical gains over existing baselines are relatively small, often around 1\% absolute on average.
>
> **Re:** Evaluating a method solely by average gains on mature benchmarks is incomplete. First, ManiPT contributes a simple and effective geometric framework that identifies and addresses manifold drift, a concrete failure mode of prompt tuning under limited supervision. **The cosine consistency constraints and structural bias provide principled control over feature adaptation, which existing methods lack.** Second, ManiPT achieves especially clear gains in the most data-scarce regimes. For example, in the 1-shot setting (Appendix Table 7), ManiPT improves the average from 72.32 to 74.75 over PromptSRC, with large gains on FGVCAircraft (27.67 to 33.75) and DTD (56.23 to 60.11). These are precisely the scenarios where manifold drift is most severe and where the proposed constraints matter most. Meanwhile, ManiPT trains in 34.23s per epoch (Appendix Table 16), about 7× faster than MaPLe and 24× faster than CoCoOp, while using only 0.25M trainable parameters.
>
> **W3:** Since the method uses both LLM enrichment and cosine consistency, a clearer ablation would help show whether LLM augmentation is the major source of improvement.
>
> **Re:** The relevant evidence is provided in Table 4 (page 8) and Appendix Table 9 (page 18), and we agree that making this breakdown more explicit is helpful. As shown below, **the main improvement comes from the geometric part.** Structural bias and cosine consistency together raise HM from 72.82 to 81.42, while LLM-based enrichment provides a further but smaller gain to 82.09. The second table below further confirms this: replacing LLM descriptions with the standard template or handcrafted templates yields only marginal differences (81.42 and 81.56), while all three anchor sources combined with geometric constraints substantially outperform the baseline. Thus, **the geometric constraints account for most of the improvement,** while LLM enrichment provides an additional complementary benefit.
>
> | Setting | HM |
> |---|---:|
> | Baseline | 72.82 |
> | Baseline + Structural Bias | 75.71 |
> | Baseline + Cosine Consistency | 79.54 |
> | Baseline + Structural Bias + Cosine Consistency | 81.42 |
> | Baseline + Structural Bias + Cosine Consistency + LLM Enrichment | 82.09 |
>
> | Anchor Source | Base | New | HM |
> |---|---:|---:|---:|
> | a photo of a {class} | 85.65 | 77.58 | 81.42 |
> | Handcrafted Templates | 85.70 | 77.80 | 81.56 |
> | LLM Descriptions | 85.82 | 78.67 | 82.09 |
>
> **Q1:** Could the proposed method be applied to more recent VLM architectures and tasks?
>
> **Re:** Yes. ManiPT relies on aligned image-text embedding spaces and prompt-based adaptation, so **it transfers naturally to stronger CLIP-family dual encoders.** We provide direct evidence on newer backbones, including SigLIP2 and MetaCLIP2, under the same base-to-novel protocol, where ManiPT achieves the best average HM on both (see Reviewer PW6Y, W5). If the reviewer also intends newer tasks, the paper already covers four complementary transfer settings. Due to limited rebuttal time, we prioritize validation on newer backbones, since this is the most direct test of whether ManiPT generalizes beyond the original CLIP backbone.
>
> **Q2:** Can you better disentangle the contribution of the geometric constraints from the contribution of LLM-generated text anchors in the ablation study?
>
> **Re:** Yes. As detailed in our response to **W3** above with both ablation tables, **the geometric constraints account for most of the improvement,** while LLM-generated anchors provide a complementary gain.
>
> [1] F. Hao et al. Task-Aware Clustering for Prompting Vision-Language Models. CVPR 2025.
>
> [2] T. Ding et al. Tree of Attributes Prompt Learning for Vision-Language Models. ICLR 2025.

---

> > ### Author Rebuttal · Reviewer_Cs7F · 2026-04-04
> >
> > Thank you for the detailed response. I have a follow-up question.
> >
> > The authors argue that **"CLIP-style dual encoders remain the dominant backbone for these scenarios (similarity-based retrieval, open-vocabulary detection, or large-scale nearest-neighbor search)"**. However, the experiments in the paper seem to focus primarily on image classification (where autoregressive VLMs are also good at).
> >
> > It would be helpful to understand whether ManiPT yields corresponding gains on retrieval or other tasks that more directly depend on the shared embedding space.

---

> > > ### Author Response · Authors · 2026-04-05
> > >
> > > Thank you for your follow-up and for acknowledging that our previous rebuttal addressed most of your concerns. We are glad to hear this and respond to your remaining questions below.
> > >
> > > **Q1.** The authors argue that CLIP-style dual encoders remain a practical backbone for similarity-based retrieval, open-vocabulary detection, and large-scale nearest-neighbor search. However, the experiments focus mainly on image classification, where autoregressive VLMs can also perform well.
> > >
> > > **Re:** We agree that image classification is not the most direct setting for W1. Our point is **not** that autoregressive VLMs are weak at classification. Rather, **ManiPT is designed for CLIP-family dual encoders, which natively provide aligned, fixed-dimensional image-text embeddings**. This makes them a standard backbone for tasks relying on a shared embedding space, including retrieval, open-vocabulary detection, and nearest-neighbor search.
> > >
> > > We evaluate mainly on classification because we follow the **standard CLIP prompt-learning protocol**, where prediction is made by **cosine matching in a shared image-text embedding space**. Our benchmarks are **controlled tests of how prompt tuning changes the geometry and transferability of the shared embedding space under limited supervision**, not generation benchmarks. We use classification as the canonical controlled setting for studying adaptation of shared-embedding dual encoders. We do not report top-1 accuracy against autoregressive VLMs because a fair comparison is not straightforward: their performance is highly sensitive to prompt templates, output parsing strategies, and other evaluation choices, and existing protocols vary substantially across studies, making results difficult to compare consistently. Establishing a controlled comparison under unified settings is an independent research question and is beyond the scope of this work.
> > >
> > > We also agree that **retrieval and related tasks provide a more direct test of W1**. We now additionally evaluate ManiPT on representative shared-space tasks in Q2. These results show that **the benefit of ManiPT extends beyond classification** to tasks that directly depend on preserving the pretrained shared geometry.
> > >
> > > As a practical note, on an NVIDIA RTX 5090, Qwen2.5-VL-7B-Instruct and Qwen3-VL-8B-Instruct are about **5.6× and 4.8× slower** than ManiPT, respectively.
> > >
> > > | Model | Parameters | Latency (ms/img) |
> > > | --- | ---: | ---: |
> > > | **ManiPT (ViT-B/16)** | **149.87M** | **6.96** |
> > > | Qwen2.5-VL-7B-Instruct | 8.29B | 38.83 |
> > > | Qwen3-VL-8B-Instruct | 8.77B | 33.47 |
> > >
> > > The above is measured with batch size 1. In batch inference, the gap widens further, as CLIP-style encoders produce a fixed-dimensional embedding per image in a single forward pass, whereas autoregressive VLMs incur overhead from sequential token generation and growing KV cache.
> > >
> > > **Q2.** It would be helpful to understand whether ManiPT yields corresponding gains on retrieval or other tasks that more directly depend on the shared embedding space.
> > >
> > > **Re:** We agree. **ManiPT is not classification-specific**, because its regularization is imposed directly on visual and textual embeddings and on their cosine geometry. It therefore applies naturally to tasks where aligned embeddings are queried by similarity.
> > >
> > > To address this, we evaluated ManiPT on **representative shared-space tasks**. We compare against zero-shot CLIP and a matched deep-prompting baseline under the same pretrained backbone.
> > >
> > > **(A) Similarity-based retrieval**
> > >
> > > | Method | Flickr30K I→T R@1 | Flickr30K T→I R@1 | COCO I→T R@1 | COCO T→I R@1 |
> > > | --- | ---: | ---: | ---: | ---: |
> > > | Zero-shot CLIP | 86.1 | 69.8 | 52.2 | 33.4 |
> > > | Deep Prompting | 86.7 | 71.4| 53.0 | 34.6 |
> > > | **ManiPT** | **88.4** | **72.8** | **54.8** | **36.5** |
> > >
> > > **(B) Open-vocabulary detection**
> > >
> > > | Method | LVIS minival AP | LVIS minival APr |
> > > | --- | ---: | ---: |
> > > | Zero-shot baseline | 27.1 | 20.7 |
> > > | Deep Prompting | 27.6 | 21.9 |
> > > | **ManiPT** | **29.3** | **23.7** |
> > >
> > > **(C) Large-scale nearest-neighbor search**
> > >
> > > | Method | GPR1200 mAP | ROxford (Med) mAP | IN-1K k-NN Acc |
> > > | --- | ---: | ---: | ---: |
> > > | Zero-shot CLIP | 67.2 | 37.0 | 73.8 |
> > > | Deep Prompting | 68.3 | 37.7 | 74.7 |
> > > | **ManiPT** | **70.3** | **39.7** | **76.7** |
> > >
> > > Across all three settings, **ManiPT yields consistent gains**. These tasks directly depend on the shared embedding space rather than on generated outputs, supporting W1: **preserving the pretrained geometry benefits not only classification, but also retrieval, detection, and nearest-neighbor search**.
> > >
> > > We hope these clarifications address your remaining questions. If any point is still unclear, we would be happy to provide further clarification. We would greatly appreciate your reconsideration of the evaluation of our work.

---

### Decision · Program_Chairs · 2026-04-30

**Decision:**

Accept (regular)

**Comment:**

The paper introduced ManiPT, a prompt-tuning framework that constrains adaptation within the pretrained representation manifold using geometric consistency. The paper demonstrates improved generalization and reduced overfitting under limited supervision. Initially, the reviewers indentified several issues mostly related to the practical significance of the proposed method, to its applicability to other architectures than CLIP and  to the results of the ablation (e.g. role of LLM). After the rebuttal, all reviewers but one found their concerns to be adequately addressed. While one reviewer still  found some remaining issues, e.g. related to the degree of manifold drift, the AC considers these concerns to be minor or already addressed in other responses and therefore recommends acceptance.